# On the Expressive Power and Limitations of Multi-Layer SSMs

## Abstract

We study how depth, finite precision, state dimension, and chain-of-thought (CoT) affect the expressive power of multi-layer state-space models (SSMs). For the explicit-table $K$-function-composition problem, a canonical benchmark for sequential information propagation, we prove that any $L$-layer SSM solving $(L + 3)$-function composition must satisfy $d^2p = \Omega(N/L^3)$, where $d$ is the state dimension and $p$ is the per-scalar precision. Conversely, $K$-function composition is solved exactly by a $(K + 1)$-layer generalized SSM with $d = 1$ and $p = \Theta(\log N)$. This gives a worst-case depth hierarchy for this formal problem family. We then distinguish post-input reasoning, in which all thought tokens are generated after the input, from input-interleaved reasoning, in which thought tokens may be inserted while the input stream is being read. Post-input reasoning does not circumvent our communication-based lower-bound pipeline, whereas input-interleaved reasoning admits bidirectional simulations with general deterministic one-pass streaming algorithms at the granularity of persistent memory. Finally, width and precision are not interchangeable under exact step-preserving simulation in the base affine-state model, but become interchangeable through the streaming-memory characterization once input-interleaved reasoning is allowed.

## 1 Introduction

State-space models (SSMs) have emerged as a promising alternative to transformers for sequence modeling, offering linear-time inference and principled mechanisms for capturing long-range dependencies (Gu et al., 2022; Gu & Dao, 2024; Dao & Gu, 2024). Architectures such as S4 (Gu et al., 2022) and Mamba (Gu & Dao, 2024) process sequences through a recurrence that is *linear* in the hidden state yet *input-dependent* in its transition parameters, enabling efficient parallel training via associative scans while retaining the streaming efficiency of recurrent models. These models have achieved strong empirical performance across language, audio, and genomics, and their multi-layer variants are now deployed at scales comparable to transformer-based large language models.

Despite this practical success, a rigorous understanding of the *expressive power* of multi-layer SSMs remains incomplete. A growing body of theoretical work has begun to map the computational landscape of these architectures. Merrill et al. (2024) showed that, under standard complexity-theoretic assumptions, common SSMs like S4 and Mamba cannot express computations outside $\mathsf{TC}^0$, placing them on a similar footing to transformers in terms of circuit complexity. Sarrof et al. (2024) studied SSM expressiveness through the lens of formal languages, identifying both strengths and weaknesses relative to transformers. Muca Cirone et al. (2024) provided a continuous-time analysis via rough path theory, characterizing the closure of linear controlled differential equations that underpin selective SSMs, while Zubic & Scaramuzza (2025b) studied regularity and stability properties of selective SSMs with discontinuous gating. While these results offer important insights, they primarily address *single-layer* or *time-invariant* models, or operate in asymptotic regimes that do not directly capture the interplay between *depth*, *finite precision*, and *state dimension* in multi-layer architectures.

In parallel, the role of *chain-of-thought* (CoT) reasoning has led to numerous theoretical works in the context of transformers. Merrill & Sabharwal (2024) showed that allowing a transformer decoder to generate intermediate tokens before answering can fundamentally expand its computational power: a linear number of CoT steps enables simulation of arbitrary finite automata, while polynomial steps yield the full power of P. Li et al. (2024) proved analogous results for constant-precision transformers, connecting CoT to circuit size. Chen et al. (2025) studied the computational power of Transformers without CoT and the function composition problem. These findings raise a natural question for SSMs: *does CoT similarly amplify the power of SSMs, and if so, does the timing of CoT generation matter?*

**This work.** We provide a unified theoretical analysis of the expressive power and limitations of multi-layer SSMs, organized around three axes: *compositional lower bounds*, *the role of CoT*, and *width–precision tradeoffs*. Our results are summarized as follows:

(i) **Lower bound via communication complexity (Theorem 1).** Under the canonical blockwise explicit-table encoding, we show that any $L$-layer SSM solving the $(L+3)$-function-composition problem must satisfy $d^2 p = \Omega(N/L^3)$, where $d$ is the state dimension, $p$ is the per-scalar precision, and $N$ is the problem size. The proof introduces a *forward communication model* (Definition 3) that captures the layer-by-layer information flow in multi-layer SSMs, and reduces to pointer chasing lower bounds (Nisan & Wigderson, 1993; Mao et al., 2025). This establishes a worst-case resource separation on a canonical sequential-composition benchmark. It is not a claim that every compositional task admits the same reduction or lower bound.

(ii) **Complementary upper bound and depth hierarchy.** We complement the lower bound with a construction showing that $K$-fold function composition can be solved exactly by a $(K+1)$-layer SSM with $d = 1$ and $p = \Theta(\log N)$, yielding $dp = O(\log N)$. Specializing to $K = L + 3$ gives a constant-gap depth hierarchy: the $(L+3)$-composition problem is easy for $L+4$ layers, hence for $O(L)$ layers, but hard for $L$ layers.

(iii) **Post-input reasoning does not remove the lower bound. Input-interleaved reasoning yields streaming equivalence (Proposition 1, Theorem 4).** We distinguish *post-input reasoning* (offline CoT), in which all self-generated tokens appear after the complete exogenous input, from *input-interleaved reasoning* (online CoT), in which such tokens may be inserted between exogenous tokens. Post-input reasoning is local post-processing in our communication reduction and therefore leaves this lower-bound pipeline unchanged. This is not a claim that post-input computation is useless for every task. In contrast, within the deterministic generalized model, input-interleaved reasoning admits bidirectional simulation with general one-pass streaming algorithms at the granularity of persistent memory. Consequently, a single-layer generalized SSM solves arbitrary-length function composition with $dp = O(\log N)$ using one thought token per exogenous token (Corollary 2).

(iv) **Width and precision are not interchangeable in the base model (Theorems 5 and 6).** We prove that in the base (no-CoT) model, the product $dp$ is *not* a complete invariant of computational power: a width-$w$, precision-$p$ machine cannot, in general, be simulated by a width-1, precision-$pw$ machine, nor vice versa. The proofs are algebraic, exploiting a counting argument over affine transition maps (Theorem 5) and the nonexistence of order-8 affine permutations over $\mathbb{F}_2^3$ (Theorem 6). However, once online CoT is allowed, the correct invariant becomes the total persistent memory $Lwp$, and width and precision become fully interchangeable (Proposition 2).

**Techniques.** Our lower bound strategy connects SSMs to multi-party communication via a *forward communication model* in which $K$ players, each holding one block of the input, communicate in $L$ synchronous rounds. The key observation (Lemma 1) is that because each SSM layer performs an *affine* state update, the effect of an entire input block on the hidden state can be summarized by a $(d^2+d)$-parameter affine map, which any downstream player can compose with its own summary. This reduction converts an $L$-layer SSM into an $L$-round protocol with message length $O(d^2 p)$. An improved serialization of these synchronous rounds into $L + 1$ alternating two-party messages, combined with the pointer chasing lower bound of Mao et al. (2025), yields the desired $\Omega(N/L^3)$ bound.

For the CoT results, the decisive issue is when thought tokens are generated relative to the exogenous input stream. Post-input reasoning amounts to local post-processing by the final player and cannot alter the information exchanged during the communication rounds. Input-interleaved reasoning permits state-dependent feedback before the next exogenous token arrives, enabling a bidirectional simulation with general deterministic one-pass streaming algorithms at the granularity of persistent memory.

**Organization.** Section 2 surveys related work. Section 3 introduces the generalized multi-layer SSM and the communication model. Section 4 presents the main lower bound and its proof strategy via the communication reduction. Section 5 gives the complementary upper bound via explicit construction. Section 6 formalizes post-input (offline) and input-interleaved (online) CoT and establishes their contrasting effects on expressiveness. Section 7 analyzes width–precision tradeoffs. Section 8 discusses implications and open problems. Full proofs are deferred to Appendices A–D. The main text retains theorem statements and proof roadmaps.

## 2 Related Work

**State-space models and efficient recurrences.** The structured state-space model S4 (Gu et al., 2022) demonstrated that linear recurrences with carefully parameterized transition matrices capture long-range dependencies while admitting efficient parallel training. This paradigm has since expanded into a rich family of architectures: S5 (Smith et al., 2023) simplifies to a diagonal parameterization purely in the time domain while retaining parallelism, the Linear Recurrent Unit (Orvieto et al., 2023) further distills the design to its minimal components, Mamba (Gu & Dao, 2024) introduces input-dependent (selective) gating and achieves transformer-competitive language modeling, S7 (Soydan et al., 2024) makes S5 input-dependent, and Mamba-2 (Dao & Gu, 2024) reveals a formal duality between selective SSMs and structured attention. Additional efficient recurrent alternatives include RWKV (Peng et al., 2023), Griffin (De et al., 2024) and GG-SSMs (Zubic & Scaramuzza, 2025a). A common structural feature of many of these models, and the only such feature used by our communication reduction, is that the update is affine in the previous hidden state once the current layer input is fixed. These affine updates can be composed over a token block. Definition 1 isolates this property while deliberately abstracting away architecture-specific parameterizations, gating constraints, training dynamics, and hardware implementations. It is therefore a common upper-level model for lower-bound analysis.

**Expressiveness and limitations of recurrent and state-space models.** The theoretical study of recurrent architectures has a long history. Siegelmann & Sontag (1995) established that recurrent neural networks with infinite-precision rational weights are Turing-complete, but this result relies crucially on unbounded precision. Under finite or saturated precision, the computational power contracts sharply: Weiss et al. (2018) showed empirically that finite-precision RNNs behave as finite automata, and Merrill et al. (2020) formalized this by proving that saturated RNNs recognize exactly the regular or counter languages depending on the gating architecture. For SSMs specifically, Merrill et al. (2024) proved that standard architectures, including S4, Mamba, and RWKV, under log-precision constraints have their outputs computable in the circuit class $\mathsf{TC}^0$. Sarrof et al. (2024) characterized SSM expressiveness through formal languages, identifying separations from transformers in both directions. On the empirical side, Deletang et al. (2023) systematically tested neural architectures against the Chomsky hierarchy, Arora et al. (2024) showed that SSMs struggle with associative recall compared to attention, Jelassi et al. (2024) demonstrated a significant gap on copying tasks, Zubic et al. (2025) showed that single-layer SSMs struggle on the function composition task, and Wen et al. (2024) identified in-context retrieval as a key bottleneck separating RNNs from transformers.

Our work departs from these prior results in two respects. First, whereas the above characterize SSMs in terms of complexity or language class membership (e.g., $\mathsf{TC}^0$, regular languages), we provide *quantitative* lower bounds on the product $d^2p$ as a function of the task parameter $N$ and the layer count $L$. Second, our analysis is inherently *multi-layer*: we show how the number of layers creates a hierarchy for compositional tasks, a phenomenon that single-layer analyses cannot capture.

**Depth separations in neural sequence models.** Depth separation is a recurring theme in neural network theory. For feedforward networks, Telgarsky (2016) proved that depth yields exponential gains in expressiveness over width. For transformers, Merrill et al. (2022) showed that constant-depth transformers under saturated precision compute exactly the functions in $\mathsf{TC}^0$. Our work establishes an analogous depth hierarchy for SSMs, but through a different proof strategy: rather than circuit simulation arguments, we reduce to multi-round communication complexity via an autoregressive protocol model. The $K$-function composition task we employ is a natural benchmark for sequential depth, closely related to the iterated function problems in the transformer and circuit complexity literature (Merrill & Sabharwal, 2023).

**Communication complexity and pointer chasing.** Multi-round communication complexity provides our primary technical tool. Nisan & Wigderson (1993) initiated the study of round–communication tradeoffs for the pointer chasing problem. Subsequent refinements by Ponzio & Radhakrishnan (1999) and Yehudayoff (2020) tightened the bounds. Most recently, Mao et al. (2025) proved the strongest known lower bound in the form of $\Omega(N/K + K)$ via a gadgetless lifting technique, which we use as a black box. The use of communication complexity to derive space and streaming lower bounds is classical (Alon et al., 1999; Kushilevitz & Nisan, 1997). Our contribution is a new *forward communication model* (Definition 3) designed to match the information flow in multi-layer SSMs. In this model, $K$ players hold consecutive input blocks and communicate over $L$ synchronous rounds that mirror the $L$ layers of the SSM. The key structural insight is that the affine recurrence in each SSM layer allows a player to compress its block's effect into an $O(d^2p)$-bit affine summary, enabling a faithful simulation by a communication protocol.

**Chain-of-thought reasoning.** Chain-of-thought (CoT) prompting (Wei et al., 2022) and the related scratchpad mechanism (Nye et al., 2021) have been shown empirically to improve the reasoning capabilities of large language models. On the theoretical side, Feng et al. (2023) showed that intermediate reasoning steps enable transformers to solve problems beyond their base expressiveness, Merrill & Sabharwal (2024) proved that bounded-precision transformer decoders with linearly many CoT steps simulate arbitrary finite automata, and with polynomially many steps capture all of $\mathsf{P}$. Li et al. (2024) connected CoT length to circuit size, and Huang et al. (2025) studied the learnability and length generalization of CoT reasoning.

These theoretical results all concern *transformers*. To our knowledge, our work provides the first formal analysis of the CoT for *SSMs*. We identify a qualitative distinction between *post-input* (offline) CoT (thought tokens generated only after the full input) and *input-interleaved* (online) CoT (thought tokens interleaved during the input stream) that exploits the sequential, autoregressive nature of SSMs. Post-input CoT amounts to post-processing by the last player in the communication model and cannot circumvent our lower bounds (Proposition 1). Input-interleaved CoT fundamentally alters the information flow, allowing feedback from deeper layers to reach earlier stream positions and collapsing the model to a universal one-pass streaming simulator (Theorem 4). This online–offline dichotomy has no direct counterpart in the transformer CoT literature, where attention over all prior positions renders the distinction less consequential.

**Width, precision, and resource tradeoffs.** The interplay between state dimension and bit precision is implicit in finite-precision analyses of recurrent models (Merrill et al., 2024; Weiss et al., 2018), where the effective capacity is governed by the total memory budget. However, to our knowledge, the question of whether width and precision are *interchangeable*, i.e., whether a width-$w$, precision-$p$ machine can always be replaced by a width-1, precision-$wp$ machine, has not been formally investigated. Our algebraic results (Theorems 5 and 6) show that in the base affine-state model the answer is negative in both directions, due to the richer algebraic structure of matrix-valued (as opposed to scalar) affine transitions. This non-interchangeability is resolved when online CoT is allowed, as the equivalence to one-pass streaming makes only the total bit budget relevant (Proposition 2).

## 3 Preliminaries

We work with the following generalized multi-layer state space model (SSM).

**Definition 1** (Generalized multi-layer SSM)**.** *Fix $L \in \mathbb{N}$ and state dimension $d \in \mathbb{N}$. Let $\{x_t\}_{t \geq 1} \subseteq \mathbb{R}^m$ be an input sequence and set $y_{0,t} = \mathrm{emb}(x_t, t)$, where* $\mathrm{emb}$ *is any fixed function. An $L$-layer SSM consists,*

*for each layer $\ell \in \{1, \ldots, L\}$ and each time $t \geq 1$, of matrices $A_{\ell,t} \in \mathbb{R}^{d \times d}$ and linear maps $B_{\ell,t}$ such that $B_{\ell,t} y_{\ell-1,t} \in \mathbb{R}^d$, together with a readout map $\mathrm{out}_{\ell,t} : \mathbb{R}^d \times \mathbb{R}^m \to \mathbb{R}^m$, and hidden/output variables $h_{\ell,t} \in \mathbb{R}^d$, $y_{\ell,t} \in \mathbb{R}^m$ obeying*

$$h_{\ell,t} = A_{\ell,t} h_{\ell,t-1} + B_{\ell,t} y_{\ell-1,t}, \qquad y_{\ell,t} = \mathrm{out}_{\ell,t}(h_{\ell,t}, y_{\ell-1,t}). \tag{1}$$

*We allow $A_{\ell,t}, B_{\ell,t}$ to depend on $\ell, t$, the input length $n$, and the current layer input $y_{\ell-1,t}$. All real-valued parameters are represented in a fixed finite-precision encoding of $p$ bits per scalar. The network output at time $t$ is $y_t := y_{L,t}$.*

**Remark 1** (Typing conventions)**.** *We write that $B_{\ell,t} \in \mathbb{R}^{d \times d}$ and $y_{\ell,t} \in \mathbb{R}^m$. To ensure the update $B_{\ell,t} y_{\ell-1,t}$ is well-defined, it suffices (and is common in the theory setting) to assume either $m = d$ or $B_{\ell,t}$ has the appropriate shape $\mathbb{R}^{d \times m}$. The arguments below only use that $b_{\ell,t} := B_{\ell,t} y_{\ell-1,t}$ is a $d$-vector with $p$-bit entries, so the proof is insensitive to this choice.*

**Remark 2** (Scope of the abstraction)**.** *Definition 1 deliberately isolates the affine-in-state recurrence used by many SSM architectures: conditional on the current layer input, each token induces an affine map of the preceding hidden state. Consequently, a lower bound for this generalized model also applies to a concrete architecture whenever that architecture is faithfully represented as a subclass under the same finite-precision resource accounting.*

*The converse should not be inferred. Our upper constructions are existential constructions in the generalized model, whose input-dependent transitions, readouts, and vector-valued internal tokens need not be realizable by a particular structured parameterization such as S4 or Mamba. They also do not establish learnability, numerical robustness, scan-parallel execution, or comparable hardware cost.*

### 3.1 Function composition task and communication models

**Definition 2** (*K*-function composition under the canonical stream encoding)**.** *Fix $N, K \in \mathbb{N}$. An instance consists of an initial element $a \in [N]$ and functions $f_1, f_2, \ldots, f_K : [N] \to [N]$. Unless stated otherwise, we use the canonical blockwise explicit-table encoding of length $n = 1 + KN$:*

$$x_1 := a, \qquad x_{1+(i-1)N+j} := f_i(j),$$

*for every $i \in \{1, \ldots, K\}$ and $j \in [N]$. Thus, the stream first presents $a$ and then lists the lookup tables of $f_1, \ldots, f_K$ consecutively, each in the order $f_i(1), \ldots, f_i(N)$. Equivalently, the blocks used by the communication reduction are*

$$I_1 = [1 : N + 1], \qquad I_i = [2 + (i-1)N : 1 + iN], \quad i \in \{2, \ldots, K\}.$$

*The goal is to output*

$$(f_K \circ f_{K-1} \circ \cdots \circ f_1)(a).$$

*An algorithm solves the task with error probability at most $1/3$ if it outputs the correct value with probability at least $2/3$ over its internal randomness. Deterministic algorithms are the special case with success probability one.*

**Why function composition?**   Function composition captures the basic operation of carrying the output of one computation into the next. We do not claim that every compositional task inherits the same lower bound. Its significance is instead that it is a canonical benchmark for sequential information propagation: related structures arise in iterated state transitions, pointer chasing, repeated transformations, and multi-step algorithmic reasoning. Thus, the result identifies a broader architectural obstruction—with a fixed SSM depth, information cannot always be adaptively propagated through an arbitrarily long chain—while the precise quantitative statement remains limited to our formal model and problem family.

An arbitrary staged deterministic computation can be described abstractly as a composition of transition maps. However, converting those maps into the explicit-table representation above may require a domain exponentially larger than the original working memory, increase the input length accordingly, or fail to preserve the stream order required by our communication reduction. Consequently, our lower bound transfers to another problem only when there is a resource-preserving and stream-order-preserving reduction from function composition or pointer chasing to that problem. No universal reduction is claimed here.

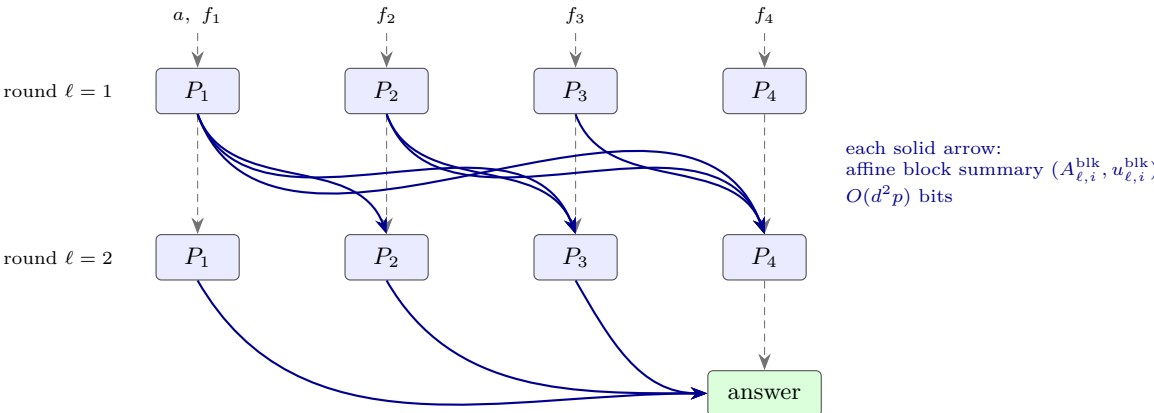

Figure 1: The forward communication model (Definition 3) for $K = 4$ players and $L = 2$ rounds. Player $i$ holds the explicit table of $f_i$ (player 1 additionally holds $a$). In each synchronous round, every player sends its affine block summary forward to all higher-indexed players; dashed arrows indicate locally stored information carried across rounds. After the final round, player $K$ outputs the answer. Round $\ell$ mirrors layer $\ell$ of the SSM in the reduction of Lemma 1.

**Definition 3** (Forward communication model). *There are $K$ players. Player $i$ holds the portion of the input corresponding to $f_i$, and player 1 additionally holds $a$. Communication proceeds in $L$ rounds (epochs). In each round $\ell \in \{1, \ldots, L\}$, every player $i$ sends a message $M_{\ell,i}$ of at most $B$ bits to all players $j > i$. Crucially, $M_{\ell,i}$ may depend on the player input and on the entire transcript of rounds $< \ell$, but not on any messages sent in the current round $\ell$ (a synchronous round). At the end of round $L$, player $K$ must output the answer.*

## 4 Lower Bounds for Multi-Layer SSMs

**Theorem 1** (Width/precision lower bound). *Fix $N, L \in \mathbb{N}$. If an $L$-layer SSM (Definition 1) solves the $(L + 3)$-function-composition problem under the blockwise explicit-table encoding of Definition 2, with error probability at most $1/3$, then*

$$d^2 p = \Omega\left(\frac{N}{L^3}\right).$$

The proof follows by combining Lemma 1 and Lemma 2.

**Proof roadmap.** The argument has four steps. *(1) Affine summaries:* once the layer input is fixed, each token induces an affine map on the hidden state, so the effect of a whole input block compresses to a $(d^2+d)$-scalar affine map. *(2) Forward protocol:* exchanging these summaries layer by layer converts an $L$-layer SSM into an $L$-round protocol with messages of $O(d^2p)$ bits (Lemma 1; see Figure 1). *(3) Serialization:* grouping the players into two super-players and serializing the $L$ synchronous rounds yields an alternating two-party protocol with $L+1$ messages (Lemma 2). *(4) Pointer chasing:* the resulting protocol solves $\mathrm{PC}_K$, so the lower bound of Mao et al. (2025) applies. Full proofs appear in Appendix A.

### 4.1 SSM $\Rightarrow$ autoregressive protocol

**Lemma 1** (Reduction from SSM to autoregressive communication). *Suppose there is an $L$-layer SSM of state dimension $d$ and precision $p$ that solves the $K$-function composition problem with error probability at most $1/3$. Then there exists an $L$-round protocol in the forward communication model (Definition 3) that solves the same problem with the same error bound and with message length $c = O(d^2p)$.*

*Proof sketch.* Once $y_{\ell-1,t}$ is fixed, each token acts on the layer-$\ell$ state as an affine map $h \mapsto A_{\ell,t}h + u_{\ell,t}$, and affine maps compose associatively over an interval. Player $i$ can therefore summarize the effect of its block on layer $\ell$ by a single pair $(A_{\ell,i}^{\mathrm{blk}}, u_{\ell,i}^{\mathrm{blk}})$ of $(d^2 + d)$ $p$-bit scalars, computable from the values $\{y_{\ell-1,t} : t \in I_i\}$ it reconstructed in the previous round. Broadcasting these summaries forward lets every player recover its incoming state $h_{\ell,s_i-1}$ exactly and reconstruct $\{(h_{\ell,t}, y_{\ell,t}) : t \in I_i\}$, which is precisely the inductive data needed for the next round. After round $L$, player $K$ holds the SSM output. Each message costs $(d^2 + d)p = O(d^2p)$ bits. The complete proof appears in Appendix A. □

### 4.2 Lower bound for autoregressive protocols

**Definition 4** (Two-party pointer chasing). *For $k \geq 1$, define the $k$-step pointer-chasing function $\mathrm{PC}_k$ : $[N]^N \times [N]^N \to \{0,1\}$ as follows. Given $f_A, f_B : [N] \to [N]$, define recursively*

$$\mathrm{pt}_0(f_A, f_B) := 1, \qquad \mathrm{pt}_r(f_A, f_B) := \begin{cases} f_A(\mathrm{pt}_{r-1}(f_A, f_B)), & r \text{ odd,} \\ f_B(\mathrm{pt}_{r-1}(f_A, f_B)), & r \text{ even,} \end{cases} \qquad r = 1, \dots, k.$$

*The output is*

$$\mathrm{PC}_k(f_A, f_B) := \mathrm{pt}_k(f_A, f_B) \bmod 2.$$

**Theorem 2** (Pointer chasing lower bound (Mao et al., 2025, Corollary 3)). *Every $(K-1)$-round randomized protocol for $\mathrm{PC}_K$ with error probability at most $1/3$ has total communication*

$$\Omega\left(\frac{N}{K} + K\right).$$

**Lemma 2.** *Let $L, K \in \mathbb{N}$ satisfy $K - L$ is odd and $K \geq L + 3$. If an $L$-round autoregressive communication protocol solves the $K$-function composition problem with error probability at most $1/3$ using messages of length at most $c$, then*

$$c = \Omega\left(\frac{N}{(L+1)K^2}\right).$$

*Proof sketch.* Setting $g_i := f_A$ for odd $i$ and $g_i := f_B$ for even $i$ turns a $\mathrm{PC}_K$ instance into a $K$-function-composition instance. Group the odd-index players into Alice and the even-index players into Bob. A synchronous round then consists of one Alice part $a_\ell$ and one Bob part $b_\ell$, each of length at most $\lceil K/2 \rceil c$. These $L$ synchronous rounds serialize into $L + 1$ alternating messages $a_1; (b_1, b_2); (a_2, a_3); \dots$ of length at most $Kc$ each, and the parity condition on $K - L$ ensures that the last speaker simulates player $K$. Applying Theorem 2 to the resulting $(K-1)$-round protocol gives $(L+1)Kc = \Omega(N/K + K)$, i.e. $c = \Omega\left(N/((L+1)K^2)\right)$. The complete proof appears in Appendix A. □

*Proof of Theorem 1.* Set $K := L + 3$. Then $K - L = 3$ is odd, so Lemma 2 applies. By Lemma 1, the assumed SSM yields an $L$-round autoregressive protocol with message length $c = O(d^2p)$. Hence

$$O(d^2p) = c = \Omega\left(\frac{N}{(L+1)K^2}\right) = \Omega\left(\frac{N}{(L+1)(L+3)^2}\right).$$

Since $(L+1)(L+3)^2 = \Theta(L^3)$, it follows that

$$d^2p = \Omega\left(\frac{N}{L^3}\right).$$

This proves the theorem. □

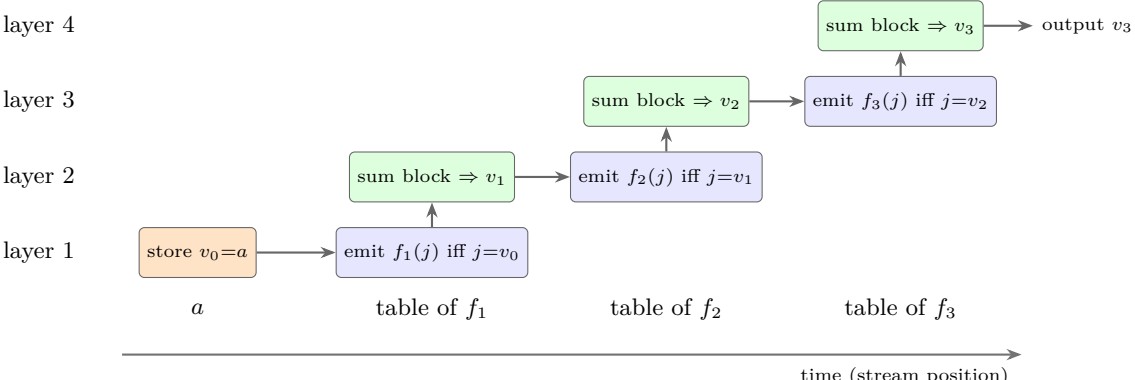

Figure 2: The gate-and-accumulate pipeline of Theorem 3, shown for $K = 3$ composed functions ($L = 4$ layers). While the table of $f_i$ streams past, layer $i$ gates the entries against its stored pointer $v_{i-1}$ and layer $i + 1$ accumulates the unique surviving value, yielding $v_i = f_i(v_{i-1})$. All layers use scalar state ($d = 1$) and $p = \Theta(\log N)$ bits of precision.

## 5 Matching Upper Bound

**Theorem 3** (Composition with logarithmic memory). *Fix $N \in \mathbb{N}$ and $L \in \mathbb{N}$ with $L \geq 2$. There exists an $L$-layer generalized SSM (in the sense of Definition 1) with state dimension $d = 1$ and finite precision $p = \Theta(\log N)$ such that, for every input $(a, f_1, f_2, \ldots, f_{L-1})$ with $a \in [N]$ and $f_i : [N] \to [N]$, the network output at the final time is exactly $f_{L-1}\big(f_{L-2}(\cdots f_1(a) \cdots)\big)$. In particular, $dp = O(\log N)$.*

*Proof sketch.* The construction is a gate-and-accumulate pipeline (Figure 2). Layer 1 loads $a$ and stores it. During the block that lists the table of $f_i$, layer $i$ passes the entry $f_i(j)$ through exactly when the within-block index $j$ matches its stored pointer $v_{i-1}$, and outputs 0 otherwise; layer $i+1$ simply sums the block, so its state after the block equals $v_i = f_i(v_{i-1})$ and is preserved thereafter. All intermediate quantities lie in $\{0, \ldots, N\}$, so $p = \Theta(\log N)$ bits of fixed-point precision suffice. The complete proof appears in Appendix B. $\quad\square$

### 5.1 Depth hierarchy

Combining the lower bound machinery of Section 4 with the upper bound of Theorem 3 yields the following depth–composition tradeoff.

**Corollary 1** (Depth–composition tradeoff for generalized SSMs). *Fix $K, L, N \in \mathbb{N}$ with $K \geq L + 3$ and $K - L$ odd. Consider the $K$-function composition problem under the row-major stream encoding*

$$x_1 := a, \qquad x_{1+(i-1)N+j} := f_i(j) \qquad \text{for } i \in \{1, \ldots, K\}, \ j \in [N].$$

*Then:*

(a) *If an $L$-layer generalized SSM of state dimension $d$ and precision $p$ solves this task with error probability at most $1/3$, then*

$$d^2 p = \Omega\left(\frac{N}{(L+1)K^2}\right).$$

(b) *There exists a $(K + 1)$-layer generalized SSM with state dimension $d = 1$ and precision $p = \Theta(\log N)$ that solves the same task exactly. In particular, $dp = O(\log N)$.*

*In particular, setting $K := L + 3$ yields a constant-gap depth hierarchy: the $(L + 3)$-function composition problem is solvable exactly by an $(L + 4)$-layer generalized SSM with $dp = O(\log N)$, whereas any $L$-layer*

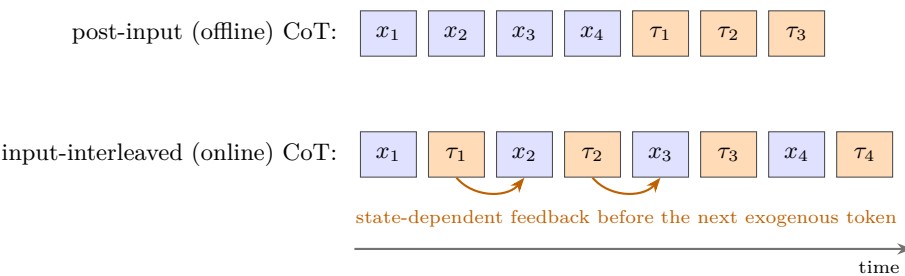

Figure 3: Token timing in the two CoT regimes. Blue boxes are exogenous input tokens, orange boxes are self-generated thought tokens. In post-input (offline) CoT, all thought tokens appear after the final exogenous token, so they cannot alter how the stream is absorbed. In input-interleaved (online) CoT, thought tokens may be inserted while the stream is being read, enabling state-dependent feedback before the next exogenous token arrives (Theorem 4).

*generalized SSM solving the same task with error probability at most* $1/3$ *must satisfy*

$$d^2 p = \Omega\left(\frac{N}{L^3}\right).$$

*Proof.* Part (a) is the direct combination of Lemma 1 and Lemma 2. Lemma 1 yields an $L$-round forward-communication protocol with message length $c = O(d^2 p)$, and Lemma 2 then implies

$$c = \Omega\left(\frac{N}{(L+1)K^2}\right).$$

Hence

$$d^2 p = \Omega\left(\frac{N}{(L+1)K^2}\right).$$

Part (b) is exactly Theorem 3 applied with its layer parameter set to $K + 1$. The final sentence is the specialization $K = L + 3$, for which

$$\frac{N}{(L+1)(L+3)^2} = \Theta\left(\frac{N}{L^3}\right). \qquad \square$$

## 6 Chain-of-Thought: Post-Input versus Input-Interleaved Reasoning

The SSM definition above assumes that the stream $(x_t)_{t=1}^n$ is exogenous. Chain-of-thought (CoT) augments this stream with self-generated tokens. We distinguish two operational regimes. In *post-input reasoning*, all self-generated tokens are produced only after the final exogenous token. In *input-interleaved reasoning*, self-generated tokens may also be inserted between exogenous tokens (Figure 3). For brevity, we refer to these regimes as *offline CoT* and *online CoT*, respectively. Here "online" describes only token timing relative to the exogenous stream. It is unrelated to online learning.

**Definition 5** (Input-interleaved (online) CoT augmentation)**.** *Fix an exogenous input stream* $(x_1, \ldots, x_n)$. *An* $L$-layer SSM with online CoT *consists of:*

- *a generalized $L$-layer SSM as in Definition 1 (with fixed finite precision p), and*

- *a* deterministic thought policy *that, after processing each exogenous token $x_i$, produces a (possibly empty) finite sequence of thought tokens $\tau_{i,1}, \ldots, \tau_{i,k_i} \in \mathbb{R}^m$, which are then fed to the SSM before the next exogenous token $x_{i+1}$ is revealed.*

*Thus the actual processed stream is*

$$x_1, \tau_{1,1}, \ldots, \tau_{1,k_1}, \ x_2, \tau_{2,1}, \ldots, \tau_{2,k_2}, \ \ldots, \ x_n, \tau_{n,1}, \ldots, \tau_{n,k_n},$$

*and the output is taken at the last processed time. We assume only that each $k_i$ is finite on every input (no uniform bound is required unless stated).*

*For an input $x_{1:n}$, define its total processed length by*

$$T(x_{1:n}) := n + \sum_{i=1}^{n} k_i(x_{1:n}).$$

*For worst-case resource statements on exogenous streams of length $n$, write*

$$T_n := \max_{x_{1:n}} T(x_{1:n}),$$

*where the maximum ranges over representable exogenous streams of length $n$. The definition requires each $k_i$ to be finite, but does not assume that $T_n$ is polynomial in $n$.*

**Definition 6** (Post-input (offline) CoT augmentation)**.** *An $L$-layer SSM with offline CoT is the special case of Definition 5 where $k_i = 0$ for all $i < n$, i.e. all thought tokens are generated only after processing $x_n$. Equivalently, the processed stream is*

$$x_1, x_2, \ldots, x_n, \tau_{n,1}, \ldots, \tau_{n,k_n},$$

*and the output is taken at the last processed time.*

### 6.1 Post-input (offline) CoT does not circumvent the communication lower-bound pipeline

Theorem 1 is the specialization $K = L + 3$ of the more general lower bound obtained by combining Lemma 1 and Lemma 2. In particular, whenever $K \geq L + 3$ and $K - L$ is odd, any $L$-layer generalized SSM solving the $K$-function composition problem must satisfy $d^2 p = \Omega\left(\frac{N}{(L+1)K^2}\right)$, and the choice $K = L + 3$ yields $d^2 p = \Omega(N/L^3)$. We show that allowing offline CoT does *not* change this implication.

**Proposition 1** (Offline CoT is local post-processing for the protocol reduction)**.** *Assume an $L$-layer generalized SSM of dimension $d$ and precision $p$ solves a streaming task (e.g. $K$-function composition) with error at most $1/3$, but is additionally allowed offline CoT steps as in Definition 6. Then there exists an $L$-round protocol in the forward communication model (Definition 3) that solves the same task with the same error bound and message length $O(d^2 p)$. Consequently, every lower bound in this manuscript obtained by combining Lemma 1 and Lemma 2 continues to hold unchanged under offline CoT. In particular, for $K$-function composition with $K \geq L + 3$ and $K - L$ odd, one still has $d^2 p = \Omega\left(\frac{N}{(L+1)K^2}\right)$, and hence $d^2 p = \Omega(N/L^3)$ when $K = L + 3$.*

*Proof sketch.* Run the reduction of Lemma 1 on the exogenous stream. After the $L$ rounds, the last player can reconstruct the entire finite-precision stack $(h_{\ell,n})_{\ell=1}^{L}$, and the post-input continuation reads no further exogenous input: all subsequent thought tokens and parameters are deterministic functions of this state and the fixed model specification. The last player therefore simulates the continuation locally, with no additional communication. The complete proof appears in Appendix C. □

**Remark 3.** *The key point is that offline CoT occurs* after *the full exogenous stream has been consumed, so it cannot alter the information transmitted during the $L$ protocol rounds. In contrast, online CoT may insert self-generated tokens* during *streaming, which can change how information propagates while the stream is still being read.*

## 6.2 Input-interleaved (online) CoT is equivalent to one-pass deterministic streaming

We now formalize the claim that online CoT renders generalized SSMs as powerful as arbitrary deterministic one-pass streaming algorithms, at the granularity of space.

**Definition 7** (Deterministic one-pass streaming algorithm). *Let $\mathcal{X}$ be the (finite-precision) input alphabet. A deterministic one-pass streaming algorithm with $S$ bits of memory is specified by: a finite memory set $\mathcal{M}$ with $|\mathcal{M}| \leq 2^S$, an initial memory state $M_0 \in \mathcal{M}$, a transition function $F : \mathcal{M} \times \mathcal{X} \to \mathcal{M}$, and an output function $G : \mathcal{M} \to \mathcal{Y}$. On input $(x_1, \ldots, x_n) \in \mathcal{X}^n$, it iterates $M_i := F(M_{i-1}, x_i)$ and outputs $G(M_n)$.*

**Operational reading of the two simulations.** Direction (A) below is a compression statement: a streaming algorithm simply carries the full finite-precision stack of layer states ($O(dpL)$ bits) plus a processed-step counter, and replays each exogenous token followed by the deterministic, finitely many thought tokens that the policy generates. Direction (B) is a serialization statement: the SSM alternates two phases per exogenous token. At the odd step it leaves its state untouched and re-emits the pair (current input, current memory) as a single thought token; at the even step, the input-dependent transition parses this token and writes the next streaming memory $F(M, x)$ into the hidden state. One thought token per exogenous token therefore suffices, and no adaptive stopping rule is needed.

**Theorem 4** (Input-interleaved (online) CoT $\Longleftrightarrow$ streaming). *Fix finite precision $p$, state dimension $d$, and exogenous input length $n$.*

*(A) (*SSM $\Rightarrow$ streaming*) Any deterministic $L$-layer generalized SSM with online CoT (Definition 5) can be simulated by a deterministic one-pass streaming algorithm whose persistent memory is*

$$O(dpL + \log T_n)$$

*bits on exogenous streams of length $n$, where $T_n$ is the machine's worst-case total processed length (Definition 5).*

*(B) (*Streaming $\Rightarrow$ SSM*) Let $\mathcal{A}$ be any deterministic one-pass streaming algorithm using $S$ bits of memory (Definition 7). Assume $dp \geq S$. Then there exists a single-layer generalized SSM with online CoT that simulates $\mathcal{A}$. The simulation inserts exactly one thought token immediately after each exogenous token (including $x_n$), and therefore uses exactly $2n$ generalized SSM steps.*

**Remark 4** (Time-index bookkeeping). *The additive $\log T_n$ term in part (A) accounts for storing an internal processed-time counter. If an external clock is supplied as read-only input, or if the relevant maps are time-homogeneous, the term can be omitted. If $T_n = \text{poly}(n)$, then $\log T_n = O(\log n)$. In particular, the construction in part (B) has $T_n = 2n$.*

*Proof sketch.* For (A), the simulator stores the stack of layer states ($O(dpL)$ bits) and an internal step counter (at most $T_n$, hence $O(\log T_n)$ bits), and replays each exogenous token followed by the deterministic, finitely many thought tokens. For (B), the SSM uses two steps per exogenous token: an odd step that re-emits ($x_i$, current memory) as a thought token without touching the state, and an even step whose input-dependent transition $B_{2i}(y) = u(y)\, e_m^\top$ writes $\text{Enc}(F(\text{Dec}(h), x_i))$ into the hidden state. Appendix C gives the full constructions. $\qquad\square$

## 6.3 Corollary: efficient iterated function composition under online CoT

**Corollary 2** (Online CoT SSMs solve function composition with logarithmic memory). *Consider the $K$-function composition problem where the stream presents the tables of $f_1, \ldots, f_K : [N] \to [N]$ in row-major order (after the initial token $a$), i.e. it enumerates the values $f_i(1), f_i(2), \ldots, f_i(N)$ for each $i$ in sequence. There exists a deterministic one-pass streaming algorithm using $O(\log N)$ bits that outputs $f_K(\cdots f_1(a) \cdots)$. Consequently, by Theorem 4(B), there exists a single-layer generalized SSM with online CoT that solves this task exactly with $dp = O(\log N)$ and with one thought token per input token.*

*Proof.* A one-pass streaming algorithm stores a current pointer $z \in [N]$, initialized as $z \leftarrow a$. At the beginning of the block encoding $f_i$, it freezes the incoming pointer by setting $q \leftarrow z$. While scanning

$$f_i(1), f_i(2), \ldots, f_i(N),$$

it maintains the within-block index $j$. When $j = q$, it stores

$$z_{\text{next}} \leftarrow f_i(j),$$

but leaves $q$ unchanged for the remainder of the table. At the end of the block, it commits

$$z \leftarrow z_{\text{next}}$$

and proceeds to the next table. Consequently, after processing table $i$,

$$z = f_i\big(f_{i-1}(\cdots f_1(a) \cdots)\big).$$

The variables $z, q, z_{\text{next}}$, and $j$ require $O(\log N)$ bits in total. The algorithm is exact, and the SSM construction follows from Theorem 4(B). $\square$

**Architectural interpretation and computational cost.** Architecturally, input-interleaved CoT can be implemented by pausing the external input stream after processing $x_i$, applying the ordinary SSM update to one or more self-generated thought tokens, and then resuming with $x_{i+1}$. When the model exposes an autoregressive feedback interface, a generated token can simply be fed back through that interface. In the particular construction of Corollary 2, exactly one thought token is inserted after each exogenous token, so no adaptive stopping mechanism is required.

For an exogenous stream of length $n$, this construction has total processed length $T = 2n$. It therefore doubles the number of recurrent updates while preserving linear update complexity and $O(\log N)$ persistent memory. This is an update-count statement, not a factor-two wall-clock guarantee: autoregressive feedback may reduce the parallelism available from associative scans, and the theorem treats its transition, readout, and thought-policy maps abstractly. Moreover, the exact construction uses finite-precision vector-valued thought tokens. It establishes architectural feasibility in the generalized model, but not an efficient discrete-token realization in a particular structured architecture such as S4 or Mamba.

## 6.4 Offline–online separation for function composition

We can now state the separation between offline and online CoT on the function composition benchmark.

**Corollary 3** (Offline–online CoT separation for function composition). *Fix $L, K \in \mathbb{N}$ with $K \geq L+3$ and $K - L$ odd. Consider the $K$-function composition problem under the row-major stream encoding*

$$x_1 := a, \qquad x_{1+(i-1)N+j} := f_i(j) \qquad \text{for } i \in \{1, \ldots, K\}, \ j \in [N].$$

*Then:*

(a) *If an $L$-layer generalized SSM with offline CoT (Definition 6) solves this task with error probability at most $1/3$, then*

$$d^2 p = \Omega\left(\frac{N}{(L+1)K^2}\right).$$

(b) *There exists an $L$-layer generalized SSM with online CoT (Definition 5) that solves the same task exactly with state dimension $d = 1$ and precision $p = \Theta(\log N)$. Equivalently,*

$$dp = O(\log N).$$

*Moreover, the construction may be chosen to use only one thought token per input token.*

*In particular, setting $K := L + 3$ yields*

$$d^2 p = \Omega\left(\frac{N}{L^3}\right)$$

*for offline CoT, whereas online CoT admits an exact construction with*

$$dp = O(\log N).$$

*Proof.* Part (a) follows from Proposition 1 together with Lemma 2, since the row-major encoding above is a valid fixed blockwise encoding for Definition 2.

For part (b), Corollary 2 gives a single-layer generalized SSM with online CoT that solves the $K$-function composition problem exactly with $dp = O(\log N)$ and one thought token per input token. This is also an $L$-layer construction after padding with $L - 1$ dummy layers that simply pass their inputs through unchanged. $\square$

## 7 Width versus Precision in Finite-Precision State-Space Machines

Now, we answer the question whether, in the generalized finite-precision SSM, a width-$w$, precision-$p$ machine can always be replaced by a width-1, precision-$pw$ machine, or conversely. The answer depends crucially on the computational model. In the base affine-state model, the product $pw$ is *not* a complete invariant: already in the one-layer case there is no universal exact simulation in either direction. By contrast, once online chain-of-thought (online CoT) is allowed, the correct invariant is total persistent memory, so a deterministic width-$w$, precision-$p$, $L$-layer machine is simulable by a width-1 machine of precision $O(Lpw + \log T_n)$, where $T_n$ is the total processed length of the simulated machine (Definition 5), and vice versa through the same streaming-memory intermediary. So, the statement "width can be traded for precision" is false in the base model and true only in the stronger online-CoT model, with the correct budget $Lpw$ rather than merely $pw$ for general $L$.

**Definition 8** (Exact step-preserving simulation). *Fix two classes of deterministic machines, both driven by the same external input stream. We say that class $\mathcal{C}'$ exactly step-preservingly simulates class $\mathcal{C}$ if, for every machine $M \in \mathcal{C}$, there exists a machine $M' \in \mathcal{C}'$ such that for every finite input stream the output produced by $M'$ at each external time step equals the output produced by $M$ at that same step. No extra external steps and no self-generated tokens are allowed.*

This is the natural interpretation of a width/precision tradeoff in the base model. Under this interpretation, the answer is negative in both directions.

### 7.1 Negative result I: width $w$ cannot, in general, be collapsed to width $1$ with precision $pw$

**Theorem 5** (No universal width-to-precision collapse in the base model). *Fix $p \geq 1$ and $w \geq 2$. In the ring model over $R_p$, the class of width-1, precision-$pw$ one-layer affine-state machines does* not *exactly step-preservingly simulate the class of width-$w$, precision-$p$ one-layer affine-state machines.*

*Proof sketch.* A universal width-$w$ machine can load an arbitrary state $x \in R_p^w$, apply an arbitrary affine map $T \in \text{Aff}(R_p^w)$ delivered as a token, and reveal the result on a common `read` token. Exact step-preserving simulation by a scalar machine forces an injective encoding of states and an injective assignment $T \mapsto F_T$ of scalar affine transitions. But $|\text{Aff}(R_p^w)| = 2^{p(w^2+w)}$ exceeds $|\text{Aff}(R_{pw})| = 2^{2pw}$ whenever $w \geq 2$, a contradiction. The complete proof appears in Appendix D. $\square$

**Remark 5.** *The proof is class-level and structural. It does not depend on any specific task lower bound. It shows directly that a one-dimensional affine state update simply has too few distinct transition maps to represent all width-w affine transitions, even when the scalar has the same total number pw of stored bits.*

### 7.2 Negative result II: the reverse collapse also fails in general

The previous theorem shows that width cannot, in general, be compressed into precision. One might hope that the reverse direction could still hold: perhaps any width-1, precision-$pw$ machine can be represented by a width-$w$, precision-$p$ machine. This is also false in general.

It suffices to exhibit one counterexample. We do so already for $(p, w) = (1, 3)$.

**Theorem 6** (No universal precision-to-width collapse in the base model). *The class of width-3, precision-1 one-layer affine-state machines over $\mathbb{F}_2^3$ does not exactly step-preservingly simulate the class of width-1, precision-3 one-layer affine-state machines over $\mathbb{Z}/8\mathbb{Z}$.*

*Proof sketch.* A width-1, precision-3 machine can load $s \in \mathbb{Z}/8\mathbb{Z}$ and then repeatedly increment modulo 8, outputting the state after each step. Any exact step-preserving simulator over $\mathbb{F}_2^3$ must realize the `inc` token by a single affine permutation of $\mathbb{F}_2^3$ of order 8. A case analysis of the possible orders of matrices in $\mathrm{GL}(3, 2)$ (Lemma 3) shows that no affine permutation of $\mathbb{F}_2^3$ has order 8, a contradiction. The complete proof appears in Appendix D. □

**Remark 6.** *Theorem 6 is a counterexample, not a complete classification. In small, low-dimensional cases, accidental equivalences can occur. The point is that there is no blanket theorem asserting that width-1, precision-pw always collapses to width-w, precision-p in the base model.*

### 7.3 What changes with online CoT

We also study a stronger model in which, between external input tokens, the machine may insert finitely many self-generated thought tokens. In that online-CoT regime, the correct invariant is no longer the pair $(w, p)$ but the total amount of persistent memory.

**Proposition 2** (Online CoT collapses width and precision to total memory). *Let $n$ be the exogenous input length, and for the machine being simulated let $T_n$ denote its worst-case total processed length (Definition 5).*

(a) *Any deterministic $L$-layer generalized SSM with online CoT, width $w$, and precision $p$, can be simulated by a deterministic single-layer generalized SSM with online CoT, width 1, and precision*

$$O(Lwp + \log T_n).$$

*Moreover, the simulation can be chosen to use one additional internal step per exogenous token.*

(b) *There exists an absolute constant $C > 0$ such that any deterministic single-layer generalized SSM with online CoT, width 1, and precision $q$, can be simulated by a deterministic single-layer generalized SSM with online CoT, width $w$, and precision $p$, provided*

$$wp \geq C(q + \log T_n).$$

*Hence, in the deterministic online-CoT model, width and precision are interchangeable up to the total persistent memory budget.*

*Proof sketch.* Both directions compose the two simulations of Theorem 4: first compress the given machine into a deterministic one-pass streaming algorithm (part (A)), then re-express that algorithm as a single-layer online-CoT SSM with the target width and precision (part (B)). The complete proof appears in Appendix D. □

**Corollary 4** (Single-layer online-CoT width/precision tradeoff). *For deterministic single-layer online-CoT machines, width $w$ and precision $p$ are interchangeable up to the product $wp$ (and the harmless $O(\log T_n)$ bookkeeping term if the processed-time index is stored explicitly). In particular, in the online-CoT model a width-$w$, precision-$p$ machine and a width-1, precision-$pw + O(\log T_n)$ machine have the same exact computational power.*

## 8 Discussion

We organize the discussion around the main themes of the paper: the depth–composition tradeoff, the role of CoT, and the width–precision landscape. We then highlight open problems.

---

**Practitioner-facing takeaways and scope.**

- **Depth and composition.** The explicit-table function-composition family exhibits a quantitative depth–memory–precision barrier. This is not a universal lower bound for all compositional tasks. A direct empirical test is to vary composition length relative to SSM depth.

- **Post-input reasoning.** Reasoning performed only after the complete input cannot circumvent the communication bottleneck used by our lower-bound pipeline. This does not mean that post-input computation is useless for every task.

- **Input-interleaved reasoning.** In the deterministic generalized model, it provides the persistent-memory power of general one-pass streaming algorithms. Corollary 2 uses one thought token per exogenous token; experiments should report internal-token, latency, and throughput overhead. This does not establish learnability, preservation of scan parallelism, or factor-two wall-clock overhead.

- **Width versus precision.** Equal bit budgets do not provide a universal exact conversion between width and precision. They should therefore be treated as separate experimental ablation axes. The theory does not rank their practical performance.

---

**Interpreting the lower bound.** Theorem 1 is the clean specialization $K = L + 3$ of our more general communication lower bound. It states that an $L$-layer SSM solving the $(L+3)$-function composition problem must satisfy $d^2 p = \Omega(N/L^3)$. For the practically relevant regime of logarithmic precision $p = \Theta(\log N)$ (i.e., each scalar is represented by $O(\log N)$ bits, as is standard in fixed-point or floating-point implementations), this becomes $d^2 = \Omega\left(\frac{N}{L^3 \log N}\right)$, requiring the state dimension to grow polynomially in $N$. This is already a strong statement: even to compose only three more functions than the layer count, an SSM cannot solve function composition over a large domain using a compact state unless its precision grows proportionally.

More generally, Corollary 1 shows that whenever $K \geq L + 3$ and $K - L$ is odd, any $L$-layer SSM solving $K$-function composition must satisfy $d^2 p = \Omega\left(\frac{N}{(L+1)K^2}\right)$. In contrast, Theorem 3 shows that $K$-fold composition can be solved exactly by a $(K+1)$-layer SSM with $d = 1$ and $p = \Theta(\log N)$. In particular, setting $K = L + 3$, the same task is solvable by an $(L+4)$-layer SSM with $dp = O(\log N)$. The gap between $L$ and $L+4$ layers is thus a genuine constant-gap depth barrier.

**Quadratic gap: $d^2 p$ versus $dp$.** The lower bound scales as $d^2 p$ while the upper bound in Section 5 achieves $dp = O(\log N)$. The factor $d^2$ arises because the affine summary communicated between blocks contains both a $d \times d$ matrix $A$ and a $d$-dimensional vector $b$, and it is the matrix component that dominates the communication cost. An individual SSM layer *stores* only $dp$ bits in its hidden state, but to *transmit the effect* of a block to a downstream player one must send the full affine map, which requires $d^2 p$ bits. Whether this gap can be closed, either by strengthening the lower bound to $dp = \Omega(\cdot)$ or by exhibiting tasks where $d^2 p$ is truly necessary, is an interesting open question. One might conjecture that for structured (e.g., diagonal or low-rank) transition matrices, the effective communication cost drops, potentially closing the gap; see the discussion of structured parameterizations below.

**Implications of the CoT dichotomy.** Post-input (offline) CoT, generating thought tokens only after the input stream has been fully consumed, provides no benefit against the communication lower-bound pipeline (Proposition 1). The intuition is clean: once the entire input has been processed, all information about the input is already compressed into the finite-precision layer states $(h_{\ell,n})_{\ell=1}^{L}$. Any post-hoc computation can only manipulate this fixed, finite summary.

Input-interleaved (online) CoT, by contrast, fundamentally restructures the computation. By inserting thought tokens *during* the input stream, it allows the model to serialize its multi-dimensional state into a scalar channel, effectively converting a multi-layer SSM into a universal one-pass streaming algorithm (Theorem 4). For function composition, this collapses the resource requirement from the offline/base lower

bound $d^2p = \Omega(N/L^3)$ in the specialization $K = L + 3$ to an exact online-CoT construction with $dp = O(\log N)$ using a single layer (Corollary 2). This has two practical implications. First, it means that the expressiveness ceiling for SSMs with online CoT is set by one-pass streaming lower bounds (e.g., $\Omega(\log N)$ bits for function composition), which are often much milder than the multi-layer lower bounds we prove for the base model. Second, it suggests that for SSM-based language models, the *timing* of intermediate reasoning steps relative to the input may be more important than their mere existence, a consideration that current CoT prompting strategies (Wei et al., 2022) for autoregressive models do not explicitly address.

We note that the online CoT model, while theoretically clean, requires a mechanism for the model to decide when and how many thought tokens to generate. In practice, this could be implemented via a "pause" or "thinking" token mechanism (Goyal et al., 2024), where the model emits special tokens that do not contribute to the output but allow internal state manipulation. Our results provide theoretical motivation for such mechanisms in SSM-based architectures. We emphasize, however, that Theorem 4 is an expressiveness and memory statement: the construction of Corollary 2 doubles the number of recurrent updates and may sacrifice associative-scan parallelism, so evaluating the accuracy–cost tradeoff of pause/thinking-token mechanisms remains an empirical direction rather than an efficiency guarantee.

**Width versus precision: interpretation.** Theorems 5 and 6 show that, in the base affine-state model, there is no universal exact, step-preserving conversion between width and numerical precision, even when the product $wp$ is held fixed. This is an algebraic non-interchangeability result; it does not establish that one resource is empirically superior to the other. In particular, the theory does not rank width and precision by predictive accuracy, learnability, numerical robustness, throughput, energy use, or hardware cost. The appropriate practical takeaway is therefore to treat state dimension and numerical precision as separate ablation axes and report both task performance and systems cost.

However, Proposition 2 shows that this non-interchangeability vanishes under online CoT, where the sole relevant quantity is the total persistent memory $Lwp$, up to the harmless $O(\log T_n)$ bookkeeping term when the processed-time index is stored explicitly. This cleanly outlines the boundary: the algebraic structure of matrix transitions matters in the base model but is neutralized once the model can serialize its state through thought tokens.

**Structured parameterizations.** Many practical SSMs use structured transition matrices, like diagonal (Gu et al., 2022; Smith et al., 2023; Orvieto et al., 2023), block-diagonal, or low-rank, rather than dense $d \times d$ matrices. In such cases, the affine summary of a block may be representable in fewer than $d^2p$ bits (e.g., $dp$ bits for a diagonal $A$). Our lower bound machinery (Lemma 1) automatically adapts: the communication cost per message equals the number of bits needed to specify the block's affine summary, which for diagonal models would be $O(dp)$ rather than $O(d^2p)$. This could lead to tighter bounds. Conversely, the upper bound construction in Section 5 already uses $d = 1$ (trivially diagonal), so the diagonal restriction does not weaken the achievable results. Fully characterizing the depth–composition tradeoff for specific parameterization families (diagonal, shift, companion, etc.) is a natural direction for future work.

**Randomized models.** Our lower bound framework accommodates randomized SSMs (via the randomized pointer chasing lower bound of Theorem 2), but the upper bound constructions are deterministic. It remains open whether randomization can provide additional power in the base SSM model, for instance, by using random projections to compress the affine summary more efficiently. In the streaming world, randomization is known to yield exponential savings for certain problems (e.g., frequency estimation (Alon et al., 1999)), and it would be interesting to determine whether analogous separations exist for SSMs.

**Learning versus expressiveness.** Our results are purely about *expressiveness*: we characterize which functions can be computed by SSMs of given dimensions, not whether gradient-based training can find the right parameters. The gap between expressiveness and learnability is well documented for transformers (Abbe et al., 2023; Barak et al., 2022) and is likely to be equally significant for SSMs. For instance, the construction in Theorem 3 uses a carefully designed readout function $\text{out}_{\ell,t}$ that performs exact index matching, a function that may be difficult to learn from data. Understanding the learnability of compositional tasks by SSMs,

and whether gradient descent on standard parameterizations can discover the constructions we exhibit, is also a direction for future work.

**Limitations and broader impact.** Our results are worst-case expressiveness statements for explicit table-encoded problems and generalized finite-precision SSMs. They do not imply that every real-world sequence task exhibits the same lower bound. Likewise, the positive constructions use hand-designed generalized transition and readout maps, and the input-interleaved construction uses vector-valued latent tokens. The results therefore do not establish learnability by gradient descent, realization in a particular S4- or Mamba-style parameterization, empirical accuracy, or wall-clock and hardware efficiency.

The work uses no human-subject data, personal data, or deployed decision-making system. Its broader impact is indirect: it may help clarify architectural resource tradeoffs and motivate more transparent reporting of reasoning-token, memory, and latency costs. A principal risk is overgeneralizing worst-case formal separations into empirical claims about all SSM tasks or implementations.

### Open problems

We conclude the discussion by listing concrete open problems suggested by our analysis.

**Q1. Closing the $d^2p$ versus $dp$ gap.** Can the lower bound for function composition be strengthened to $dp = \Omega(N/\mathrm{poly}(L))$? If not, does there exist a family of tasks $T_N$ for which $d^2p$ is the correct complexity measure in the base model, in the sense that every SSM solving $T_N$ must satisfy $d^2p = \Omega(f(N))$, while $T_N$ is solvable by some SSM with $d^2p = O(f(N))$ and $dp = o(f(N))$?

**Q2. Tightening the dependence on depth and composition length.** Our general lower bound gives $d^2p = \Omega\left(\frac{N}{(L+1)K^2}\right)$ for $K \geq L+3$ and $K-L$ odd, while the upper bound shows that $K+1$ layers suffice to compose $K$ functions exactly. Can the lower bound be sharpened, either in its dependence on $L$ and $K$ or in the constant-gap specialization $K = L+3$? For example, can the $N/L^3$ term be improved to $N/L^2$ or $N/L$?

**Q3. Diagonal and structured SSMs.** What are tight depth–composition tradeoffs when $A_{\ell,t}$ is restricted to be diagonal, shift-structured, or low-rank?

**Q4. Randomized SSMs.** Does internal randomness provably help multi-layer SSMs on compositional tasks in the base (no-CoT) model?

**Q5. Learnability of compositional constructions.** Can standard SSM training (e.g., gradient descent on Mamba-style parameterizations) learn to solve $K$-function composition when $K$ is close to the layer count, and if so, with what sample complexity?

**Q6. Online CoT token complexity.** Our online CoT constructions use one thought token per input token. Is a sublinear number of thought tokens sufficient for function composition, or is there a thought-token lower bound?

## 9   Conclusion

We have presented a theoretical analysis of multi-layer SSMs organized around three topics: depth, CoT, and resource tradeoffs. Our main result (Theorem 1) establishes that $L$-layer SSMs require $d^2p = \Omega(N/L^3)$ to solve the $(L+3)$-function composition problem, via a reduction to multi-round communication complexity through a forward communication model. More generally, our communication argument yields the bound $d^2p = \Omega\left(\frac{N}{(L+1)K^2}\right)$ for $K$-function composition whenever $K \geq L+3$ and $K-L$ is odd. A complementary upper bound shows that $(K+1)$ layers suffice with $dp = O(\log N)$ to compose $K$ functions, yielding a constant-gap depth hierarchy in the specialization $K = L+3$.

Furthermore, we formalized the distinction between post-input (offline) and input-interleaved (online) CoT for SSMs and proved a sharp separation: post-input CoT does not circumvent the communication lower-bound pipeline, while input-interleaved CoT makes multi-layer SSMs equivalent in power to deterministic one-pass streaming algorithms. This equivalence is tight in both directions and offers a clean characterization of the additional power conferred by interleaving self-generated tokens with the input stream. Finally, we showed that width and precision are *not* interchangeable resources in the base SSM model, which is a consequence of the richer algebraic structure of matrix-valued versus scalar affine transitions, but become fully interchangeable under online CoT, where only the total memory budget matters.

Taken together, these results provide a unified theoretical framework for understanding how depth, finite precision, and CoT shape the computational power of SSMs, and they suggest concrete architectural principles: (i) depth is essential for the explicit-table composition family studied here under the assumptions of our model, (ii) online reasoning tokens can substitute for depth and width, and (iii) state dimension and numerical precision play fundamentally different roles in the absence of such tokens.

## A   Deferred Proofs for the Communication Lower Bound

*Proof of Lemma 1.* Fix an instance $(a, f_1, \ldots, f_K)$ and its canonical token stream of length $n = 1 + KN$ (Definition 2). Let

$$I_1 = [1 : N + 1], \qquad I_i = [2 + (i-1)N : 1 + iN] \quad (i = 2, \ldots, K).$$

These intervals partition $[n]$; write $I_i = [s_i : e_i]$. Player 1 owns the tokens encoding $a$ and $f_1$, while player $i$ owns exactly the tokens encoding $f_i$ for $i \geq 2$.

Let $S$ be the given $L$-layer SSM. Fix a layer $\ell \in \{1, \ldots, L\}$. For each time $t$, write $u_{\ell,t} := B_{\ell,t} y_{\ell-1,t} \in \mathbb{R}^d$, so that the layer-$\ell$ state update is $h_{\ell,t} = A_{\ell,t} h_{\ell,t-1} + u_{\ell,t}$. Thus, once $y_{\ell-1,t}$ is regarded as fixed, each token induces an affine map $h \longmapsto A_{\ell,t} h + u_{\ell,t}$.

For any interval $J = [u : v] \subseteq [n]$, define the composed affine map $T_{\ell,J}(h) = A_{\ell,J} h + u_{\ell,J}$, where

$$A_{\ell,J} := A_{\ell,v} A_{\ell,v-1} \cdots A_{\ell,u},$$

and

$$u_{\ell,J} := \sum_{j=u}^{v} \Big( \prod_{r=j+1}^{v} A_{\ell,r} \Big) u_{\ell,j},$$

with the empty product interpreted as the identity. A standard induction on $|J|$ shows that if the incoming state at time $u - 1$ is $h_{\ell,u-1}$, then

$$h_{\ell,v} = T_{\ell,J}(h_{\ell,u-1}) = A_{\ell,J} h_{\ell,u-1} + u_{\ell,J}.$$

Moreover, if $J_1 = [u : v]$ and $J_2 = [v + 1 : w]$, then

$$T_{\ell,J_2 \cup J_1} = T_{\ell,J_2} \circ T_{\ell,J_1}.$$

For each block $I_i = [s_i : e_i]$, define its layer-$\ell$ summary by

$$T_{\ell,I_i}(h) = A_{\ell,i}^{\mathrm{blk}} h + u_{\ell,i}^{\mathrm{blk}}, \qquad A_{\ell,i}^{\mathrm{blk}} := A_{\ell,I_i}, \quad u_{\ell,i}^{\mathrm{blk}} := u_{\ell,I_i}.$$

We now describe the autoregressive protocol. Player $i$ holds exactly the input tokens in $I_i$.

At round $\ell$, assume inductively that player $i$ already knows the correct values $\{y_{\ell-1,t} : t \in I_i\}$. From these local values, player $i$ can instantiate all matrices $A_{\ell,t}, B_{\ell,t}$ for $t \in I_i$, compute $u_{\ell,t} = B_{\ell,t} y_{\ell-1,t}$, and hence compute the block summary $\big(A_{\ell,i}^{\mathrm{blk}}, u_{\ell,i}^{\mathrm{blk}}\big)$. It sends this pair to every player $j > i$.

After receiving the summaries from players $1, \ldots, i - 1$, player $i$ forms the prefix composition

$$P_{\ell,i-1} := T_{\ell,I_{i-1}} \circ \cdots \circ T_{\ell,I_1}, \qquad P_{\ell,0} := \mathrm{id}.$$

Since $h_{\ell,0}$ is fixed, player $i$ can recover the incoming state $h_{\ell,s_i-1} = P_{\ell,i-1}(h_{\ell,0})$. It then computes sequentially, for every $t \in I_i$,

$$h_{\ell,t} = A_{\ell,t}h_{\ell,t-1} + u_{\ell,t}, \qquad y_{\ell,t} = \text{out}_{\ell,t}(h_{\ell,t}, y_{\ell-1,t}).$$

These values are stored locally and constitute the inductive data needed for round $\ell + 1$.

The induction on $\ell$ is immediate. For $\ell = 1$, each player can compute $y_{0,t} = \text{emb}(x_t, t)$ on its own interval directly from the input stream. If the claim holds for layer $\ell - 1$, then the block summaries computed at round $\ell$ agree exactly with those of the SSM, the reconstructed incoming state is correct, and hence the locally reconstructed $(h_{\ell,t}, y_{\ell,t})$ are exactly the SSM values on that interval. After round $L$, player $K$ therefore holds exactly the SSM output, and the error probability is unchanged.

Finally, each message consists of one $d \times d$ matrix and one $d$-vector, i.e. $d^2 + d$ scalars, each represented with $p$ bits. Hence $c \leq (d^2 + d)p = O(d^2 p)$. This proves the claim. $\qquad\square$

*Proof of Lemma 2.* Let $\Pi$ be such an $L$-round autoregressive protocol. We reduce $\text{PC}_K$ to $K$-function composition. Given $f_A, f_B : [N] \to [N]$, define a composition instance by setting

$$a := 1, \qquad g_i := \begin{cases} f_A, & i \text{ odd}, \\ f_B, & i \text{ even}. \end{cases}$$

Then $g_K(g_{K-1}(\cdots g_1(1) \cdots)) = \text{pt}_K(f_A, f_B)$, so computing the composition value determines $\text{PC}_K(f_A, f_B)$.

Now group the $K$ players of $\Pi$ into two super-players: Alice simulates all odd-index players, and Bob simulates all even-index players.

For each autoregressive round $\ell \in \{1, \ldots, L\}$, let

$$a_\ell := \text{concatenation of all round-}\ell\text{ messages sent by odd players},$$

$$b_\ell := \text{concatenation of all round-}\ell\text{ messages sent by even players}.$$

Because the autoregressive model is synchronous, every round-$\ell$ message depends only on the local input and on the transcript of rounds $< \ell$. Therefore $a_\ell$ depends only on Alice's input and on $(a_1, b_1, \ldots, a_{\ell-1}, b_{\ell-1})$, and similarly $b_\ell$ depends only on Bob's input and the same prior transcript. Thus the pair $(a_\ell, b_\ell)$ forms an $L$-round simultaneous two-party protocol.

Moreover, in each round there are at most $\lceil K/2 \rceil$ odd-player messages and at most $\lfloor K/2 \rfloor$ even-player messages, so

$$|a_\ell| \leq \left\lceil \frac{K}{2} \right\rceil c, \qquad |b_\ell| \leq \left\lfloor \frac{K}{2} \right\rfloor c.$$

We now serialize this simultaneous protocol into an alternating one. The transcript is sent in the following order:

$$a_1; \ (b_1, b_2); \ (a_2, a_3); \ (b_3, b_4); \ \cdots,$$

where nonexistent terms at the end are omitted. More formally, the alternating messages are

$$m_1 := a_1,$$

$$m_{2r} := (b_{2r-1}, b_{2r}), \qquad m_{2r+1} := (a_{2r}, a_{2r+1}),$$

with missing components deleted when an index exceeds $L$.

This serialization is valid because once a party knows the simultaneous transcript through round $t - 1$ and has just computed its own round-$t$ message, it can also compute its round-$(t+1)$ message immediately after receiving the other party's round-$t$ message. Concretely:

- after Alice sends $a_1$, Bob can compute $b_1$, and then $b_2$;
- after Bob sends $(b_1, b_2)$, Alice can compute $a_2$, and then $a_3$;

- and so on.

Hence the simultaneous transcript is fully serialized in $L + 1$ alternating messages. Each such message has length at most

$$|a_\ell| + |a_{\ell+1}| \leq Kc \quad \text{or} \quad |b_\ell| + |b_{\ell+1}| \leq Kc,$$

so the total communication is $O((L + 1)Kc)$.

At the end of these $L+1$ messages, both parties know the entire simulated simultaneous transcript. The last speaker is Alice when $L$ is even and Bob when $L$ is odd. Since Alice simulates odd-index players and Bob simulates even-index players, the last speaker simulates player $K$ exactly when $K - L$ is odd, which is our assumption. Therefore the last speaker can compute the value of player $K$, namely $\text{pt}_K(f_A, f_B)$, and append the output bit $\text{pt}_K(f_A, f_B) \bmod 2$ to its final message. We have thus produced an Alice-first alternating protocol for $\text{PC}_K$ with at most $L + 1$ messages and total communication $O((L + 1)Kc)$. If necessary, we pad with empty dummy messages to obtain exactly $K - 1$ rounds. This does not change the asymptotic communication.

Since $K \geq L + 3$, we have $L + 1 \leq K - 1$. Theorem 2 therefore applies and yields

$$(L + 1)Kc = \Omega\left(\frac{N}{K} + K\right).$$

Rearranging,

$$c = \Omega\left(\frac{N/K + K}{(L + 1)K}\right) = \Omega\left(\frac{N}{(L + 1)K^2} + \frac{1}{L + 1}\right) = \Omega\left(\frac{N}{(L + 1)K^2}\right).$$

This proves the lemma. $\qquad\square$

## B  Deferred Proof of the Matching Construction

*Proof of Theorem 3.* As already said, we work in the generalized multi-layer SSM model where, for each layer $\ell \in \{1, \ldots, L\}$ and time $t \geq 1$, the hidden state and output obey

$$h_{\ell,t} = A_{\ell,t}h_{\ell,t-1} + B_{\ell,t}y_{\ell-1,t}, \qquad y_{\ell,t} = \text{out}_{\ell,t}(h_{\ell,t}, y_{\ell-1,t}),$$

and $y_{0,t} = \text{emb}(x_t, t)$ is the embedded input token stream. We will choose $m = 1$ and $d = 1$, so all quantities above are scalars.

**Encoding of the input stream.** We fix the following (canonical) stream encoding of the instance $(a, f_1, \ldots, f_{L-1})$. Let $K := L - 1$ and define the stream length

$$n := 1 + KN = 1 + (L - 1)N.$$

Define tokens $x_1, x_2, \ldots, x_n$ by

$$x_1 := a, \qquad x_{1+(i-1)N+j} := f_i(j) \quad \text{for } i \in \{1, \ldots, K\}, \ j \in \{1, \ldots, N\}.$$

Thus, after the first token, the stream lists the truth tables of $f_1, f_2, \ldots, f_K$ consecutively in row-major order. Define the embedding to be the identity $\text{emb}(x_t, t) := x_t$, so that $y_{0,t} = x_t$.

**Precision choice.** Choose a fixed-point (or integer) encoding with $p$ bits per scalar large enough to represent every integer in $\{0, 1, \ldots, N\}$ exactly. For concreteness, it suffices to take $p \geq \lceil \log_2(N + 1) \rceil + 1$. All quantities we compute below will lie in $\{0, 1, \ldots, N\}$ and hence are exactly representable at this precision.

**High-level idea.** Write

$$v_0 := a, \qquad v_i := f_i(v_{i-1}) \quad (i = 1, \ldots, K).$$

We will maintain the invariant that, after the block encoding $f_i$ has been fully read, layer $i + 1$ stores $v_i$ in its hidden state. The computation of $v_i$ is done in a two-layer pipeline: layer $i$ *gates* the table entries of $f_i$ so that exactly one nonzero value passes to layer $i + 1$, and layer $i + 1$ *accumulates* these values over the block.

**Definition of the SSM parameters.** All layers have scalar state ($d = 1$) and scalar outputs ($m = 1$). Set initial states $h_{\ell,0} = 0$ for all $\ell \in \{1, \ldots, L\}$.

Define time indices for each function block: for $i \in \{1, \ldots, K\}$, let

$$s_i := 2 + (i-1)N, \qquad e_i := 1 + iN,$$

so the $i$th function block occupies times $t \in [s_i : e_i]$ and has length $N$. For each such $t \in [s_i : e_i]$, define the within-block index

$$j_i(t) := t - s_i + 1 \in \{1, \ldots, N\},$$

so that $x_t = f_i(j_i(t))$ on that block.

**Layer 1 (store $a$).** Set

$$A_{1,1} = 0, \quad B_{1,1} = 1, \qquad A_{1,t} = 1, \quad B_{1,t} = 0 \quad \text{for all } t \geq 2.$$

Thus $h_{1,1} = y_{0,1} = a$ and thereafter $h_{1,t} = h_{1,t-1} = a$ for all $t \geq 2$.

**Layers $\ell \in \{2, \ldots, L\}$ (accumulate exactly one gated value).** For $\ell \geq 2$, define $i := \ell - 1$ (so $i \in \{1, \ldots, K\}$ when $\ell \leq L$). Set

$$A_{\ell,t} := 1 \quad \text{for all } t \geq 1,$$

and

$$B_{\ell,t} := \begin{cases} 1, & t \in [s_i : e_i], \\ 0, & \text{otherwise.} \end{cases}$$

Hence layer $\ell = i + 1$ performs the update $h_{\ell,t} = h_{\ell,t-1} + y_{\ell-1,t}$ during the $i$th block and remains constant outside that block.

**Readout maps.** For each $\ell \in \{1, \ldots, K\}$ and time $t \geq 1$, define $\text{out}_{\ell,t}$ by

$$\text{out}_{\ell,t}(h, y) := \begin{cases} y, & t \notin [s_\ell : e_\ell], \\ y, & t \in [s_\ell : e_\ell] \text{ and } h = j_\ell(t), \\ 0, & t \in [s_\ell : e_\ell] \text{ and } h \neq j_\ell(t). \end{cases}$$

In words: layer $\ell$ passes its input through unchanged at all times except during the block encoding $f_\ell$, where it outputs the current table entry iff its stored pointer equals the current index. Finally, for the last layer $L$, define

$$\text{out}_{L,t}(h, y) := h \qquad \text{for all } t \geq 1.$$

Therefore the network output at time $t$ is $y_t = y_{L,t} = h_{L,t}$.

**Correctness.** We prove by induction on $i \in \{0, 1, \ldots, K\}$ that

$$h_{i+1,t} = v_i \quad \text{for all } t \geq e_i, \tag{2}$$

where we interpret $e_0 := 1$ (so that the claim for $i = 0$ states $h_{1,t} = v_0 = a$ for all $t \geq 1$).

*Base case $i = 0$.* By construction, $h_{1,1} = a = v_0$ and for $t \geq 2$ we have $h_{1,t} = h_{1,t-1}$, so $h_{1,t} = v_0$ for all $t \geq 1 = e_0$.

*Inductive step.* Fix $i \in \{1, \ldots, K\}$ and assume equation 2 holds for $i - 1$, i.e., $h_{i,t} = v_{i-1}$ for all $t \geq e_{i-1}$. In particular, throughout the entire block $t \in [s_i : e_i]$ we have $t \geq e_{i-1} + 1$, hence

$$h_{i,t} = v_{i-1} \qquad \text{for all } t \in [s_i : e_i].$$

Next observe that for all layers $\ell < i$ and times $t \in [s_i : e_i]$, we have $t \notin [s_\ell : e_\ell]$ because the blocks are consecutive and strictly increasing in $i$. Therefore, for such $t$, the readouts satisfy $y_{\ell,t} = y_{\ell-1,t}$ for all $\ell < i$, from where

$$y_{i-1,t} = y_{0,t} = x_t = f_i(j_i(t)) \qquad \text{for all } t \in [s_i : e_i].$$

Applying the definition of $\text{out}_{i,t}$ on the $i$th block yields

$$y_{i,t} = \begin{cases} f_i(j_i(t)), & j_i(t) = h_{i,t} = v_{i-1}, \\ 0, & \text{otherwise,} \end{cases} \qquad \text{for } t \in [s_i : e_i].$$

Since $j_i(t)$ ranges over $\{1, \ldots, N\}$ exactly once as $t$ ranges over $[s_i : e_i]$, there is a unique time $t^\star \in [s_i : e_i]$ for which $j_i(t^\star) = v_{i-1}$, and at that time $y_{i,t^\star} = f_i(v_{i-1}) = v_i$, while $y_{i,t} = 0$ for all $t \neq t^\star$ in the block.

Now consider layer $i+1$. By construction, $B_{i+1,t} = 0$ for all $t < s_i$, so $h_{i+1,s_i-1} = h_{i+1,0} = 0$. For $t \in [s_i : e_i]$ we have $B_{i+1,t} = 1$ and $A_{i+1,t} = 1$, hence

$$h_{i+1,t} = h_{i+1,t-1} + y_{i,t} \qquad (t \in [s_i : e_i]).$$

Unrolling the recurrence to the end of the block gives

$$h_{i+1,e_i} = \sum_{t=s_i}^{e_i} y_{i,t} = v_i.$$

Finally, for $t > e_i$ we have $B_{i+1,t} = 0$ and $A_{i+1,t} = 1$, so $h_{i+1,t} = h_{i+1,t-1}$ and thus $h_{i+1,t} = v_i$ for all $t \geq e_i$. This proves equation 2 for $i$.

*Conclusion.* Taking $i = K = L - 1$, we have $e_K = n$, so the invariant yields

$$h_{L,n} = v_{L-1} = f_{L-1}\big(f_{L-2}(\cdots f_1(a) \cdots)\big).$$

Since $\text{out}_{L,n}(h, y) = h$, the network output at time $n$ is $y_{L,n} = h_{L,n}$, which is the desired value.

**Resource bound.** We used state dimension $d = 1$. Choosing $p \geq \lceil \log_2(N+1) \rceil + 1$ ensures all intermediate values lie in $\{0, 1, \ldots, N\}$ and are exactly representable. Hence $dp = O(\log N)$. $\qquad \square$

## C  Deferred Proofs for the CoT Results

*Proof of Proposition 1.* Fix an input instance and its associated exogenous token stream $(x_t)_{t=1}^n$ (as in Definition 2). Consider the execution of the offline-CoT SSM: it processes $(x_t)_{t=1}^n$ first, reaching some global internal state at time $n$ (consisting of all layer states $(h_{\ell,n})_{\ell=1}^L$ and any other finite-precision registers implicit in the implementation), and then performs additional offline CoT time steps using only self-generated tokens and the fixed model specification.

We construct an $L$-round protocol in the forward communication model as follows.

1. Use *exactly* the reduction of Lemma 1 to simulate the SSM *up to time $n$* on the exogenous stream. This produces, for each layer $\ell$, the correct affine block summaries and therefore allows the last player (player $K$) to reconstruct the layer-$\ell$ state at the end of the exogenous stream, i.e. $h_{\ell,n}$, by composing the received block summaries and applying the resulting affine map to the fixed initial state. The communication per message is the same as in Lemma 1, namely $O(d^2 p)$ bits.

2. After the $L$ protocol rounds, no further communication is performed. The last player now possesses (i) the full model description and (ii) the reconstructed stack of finite-precision layer states $(h_{\ell,n})_{\ell=1}^L$. Starting from this state, it can *locally* simulate the offline CoT continuation (Definition 6), because:

   - the continuation reads no further exogenous input, and

- all future thought tokens and SSM parameters are deterministic functions of the current finite-precision state and the fixed model specification.

Therefore the last player can reproduce the same final output distribution that the offline-CoT SSM would produce.

Hence we obtain an $L$-round forward-communication protocol with message length $O(d^2p)$ and the same error guarantee. Any subsequent lower bound argument that applies to such protocols, such as Lemma 2, applies verbatim, proving the claim. $\qquad\square$

*Proof of Theorem 4.* **(A) SSM $\Rightarrow$ streaming.** Fix a deterministic $L$-layer SSM with online CoT. Because all hidden-state coordinates are $p$-bit scalars, each layer state $h_{\ell,t} \in \mathbb{R}^d$ can be stored in $O(dp)$ bits, hence the entire stack $(h_{1,t}, \ldots, h_{L,t})$ in $O(dpL)$ bits.

A streaming simulator proceeds token-by-token over the exogenous stream $(x_i)_{i=1}^n$. Upon reading $x_i$, it:

1. computes $y_{0,t} = \mathrm{emb}(x_i, t)$ and updates the layer states sequentially using equation 1 to obtain the new outputs and states at that time step;

2. computes the thought policy's next thought token(s) (if any) as deterministic functions of the current finite-precision configuration;

3. feeds each thought token back through emb and applies the same state update, repeating until the thought policy halts for this $i$ (which is guaranteed to happen after finitely many steps by Definition 5).

The simulator carries forward only the current finite-precision stack state (and a counter tracking how many internal steps have occurred, which is at most the total processed length $T_n$ and hence requires $O(\log T_n)$ bits), which is $O(dpL + \log T_n)$ bits in total. This exactly reproduces the SSM's final output, hence simulates it.

**(B) Streaming $\Rightarrow$ SSM.** Let $\mathcal{A}$ be a deterministic streaming algorithm with memory set $\mathcal{M}$, $|\mathcal{M}| \leq 2^S$, transition function $F$, and output $G$. Assume $dp \geq S$. Choose an injective encoding $\mathrm{Enc} : \mathcal{M} \to \{0, \ldots, 2^p - 1\}^d \subset \mathbb{R}^d$. (For example, fix a bijection between $\mathcal{M}$ and a subset of $\{0,1\}^{dp}$ and group the $dp$ bits into $d$ blocks of $p$ bits.) Let Dec denote its inverse on $\mathrm{Enc}(\mathcal{M})$.

We construct a *single-layer* SSM (so we drop the layer index) that processes an augmented stream of length $2n$ consisting of exogenous tokens at odd times and one thought token at each even time. We choose the token dimension $m := m_x + d + 1$, where $m_x$ is the (finite-precision) dimension used to represent exogenous inputs. We assume the embedding appends a bias coordinate: $\mathrm{emb}(x, t) := (x, 0^d, 1) \in \mathbb{R}^m$.

*Odd times ($t = 2i - 1$): expose $(x_i, M_{i-1})$ as a thought token without changing state.* Set

$$A_{2i-1} = I_d, \qquad B_{2i-1} = 0.$$

Thus $h_{2i-1} = h_{2i-2}$. Define the readout map at odd times by

$$\mathrm{out}_{2i-1}(h, y) := \big(x,\, h,\, 1\big) \in \mathbb{R}^m,$$

where $x$ denotes the first $m_x$ coordinates of $y = \mathrm{emb}(x_i, 2i-1)$. Hence the layer output at time $2i - 1$ is

$$y_{2i-1} = (x_i, h_{2i-2}, 1).$$

We interpret $y_{2i-1}$ as the (single) thought token inserted before $x_{i+1}$.

*Even times ($t = 2i$): update the hidden state to the next streaming memory.* At time $2i$, the current token is the thought token $y_{0,2i} = y_{2i-1}$. Set

$$A_{2i} = 0,$$

and define a *matrix-valued* function $B_{2i}(\cdot)$ on the set of representable inputs $y \in \mathbb{R}^m$ by

$$B_{2i}(y) := u(y) \, e_m^\top, \qquad u(y) := \mathrm{Enc}\Big(F\big(\mathrm{Dec}(h), x\big)\Big) \in \mathbb{R}^d,$$

where $y = (x, h, 1)$ is parsed into its $m_x$-coordinate input part $x$, its $d$-coordinate state part $h$, and its bias coordinate 1, and $e_m$ is the $m$th standard basis vector. Because the last coordinate of $y$ equals 1, we get

$$h_{2i} \;=\; A_{2i} h_{2i-1} + B_{2i}(y) \, y \;=\; u(y) \, (e_m^\top y) \;=\; u(y).$$

Therefore $h_{2i} = \mathrm{Enc}\Big(F\big(\mathrm{Dec}(h_{2i-2}), x_i\big)\Big)$. Initialize the SSM with $h_0 := \mathrm{Enc}(M_0)$. An induction on $i$ shows $h_{2i} = \mathrm{Enc}(M_i)$ for all $i \in \{0, 1, \ldots, n\}$.

Finally, define the readout at the last time $2n$ to emit the streaming output:

$$\mathrm{out}_{2n}(h, y) := \mathrm{Enc}_{\mathcal{Y}}\big(G(\mathrm{Dec}(h))\big),$$

where $\mathrm{Enc}_{\mathcal{Y}}$ is any fixed embedding of the output alphabet $\mathcal{Y}$ into $\mathbb{R}^m$ (e.g. store it in the first coordinate and zero elsewhere). Then the SSM output at time $2n$ equals $\mathcal{A}$'s output on the exogenous stream.

All quantities involved are finite-precision: the state $h_t$ always lies in the representable set $\mathrm{Enc}(\mathcal{M}) \subset \{0, \ldots, 2^p - 1\}^d$, and $B_{2i}(y)$ has entries in $\{0, \ldots, 2^p - 1\}$ because $u(y)$ does. The construction inserts exactly one thought token after each exogenous token (including $x_n$), so the total processed length is exactly $2n$. This completes the construction. $\qquad\square$

## D  Deferred Proofs for the Width–Precision Results

*Proof of Theorem 5.* Let $\mathcal{A}_{w,p} := \mathrm{Aff}(R_p^w) = \{h \mapsto Ah + b : A \in R_p^{w \times w}, \; b \in R_p^w\}$ be the full set of affine self-maps of $R_p^w$. Consider the following width-$w$, precision-$p$ one-layer machine $U_{w,p}$.

*Input alphabet.* The first token is an element $x \in R_p^w$. The second token is an affine map $T = (A, b) \in \mathcal{A}_{w,p}$. The third token is a fixed symbol `read`.

*Dynamics.*

- At time $t = 1$, the machine loads the hidden state with the input vector: $h_1 = x$. This is realized by choosing $A_{1,x} = 0$ and $b_{1,x} = x$.

- At time $t = 2$, upon reading $T = (A, b)$, the machine applies $h_2 = Ah_1 + b$, and produces no relevant output.

- At time $t = 3$, upon reading the common token `read`, the machine leaves the state unchanged and outputs the current hidden state.

Thus, on input $(x, T, \texttt{read})$, the output at time 3 is exactly $T(x)$.

Assume, for contradiction, that there exists a width-1, precision-$pw$ one-layer affine-state machine $V$ that exactly step-preservingly simulates $U_{w,p}$.

Let $S := R_{pw}$ be the scalar state space of $V$. For each $x \in R_p^w$, let $s_x \in S$ denote the state of $V$ after reading the first token $x$.

We first claim that the map

$$E : R_p^w \to S, \qquad E(x) := s_x,$$

is injective. Indeed, if $E(x) = E(x')$ for some $x \neq x'$, then on the common suffix $(\mathrm{id}, \texttt{read})$ the simulator would be in the same scalar state at times 2 and 3 for both inputs $(x, \mathrm{id}, \texttt{read})$ and $(x', \mathrm{id}, \texttt{read})$, hence would produce the same output at time 3. But $U_{w,p}$ outputs $x$ on the first input and $x'$ on the second, contradiction. Since $|R_p^w| = 2^{pw} = |R_{pw}|$, this injective map is in fact bijective.

For each affine map $T = (A, b) \in \mathcal{A}_{w,p}$, let

$$F_T(s) = \alpha_T s + \beta_T \qquad (\alpha_T, \beta_T \in R_{pw})$$

be the scalar affine map implemented by $V$ at time 2 on input token $T$.

We next claim that the assignment $T \mapsto F_T$ is injective. Suppose $F_T = F_{T'}$ for two affine maps $T \neq T'$. Since $T$ and $T'$ are distinct functions on $R_p^w$, there exists $x \in R_p^w$ with $T(x) \neq T'(x)$. The simulator, started from the first token $x$, reaches the same scalar state after time 2 on the input $(x, T, \texttt{read})$ as on $(x, T', \texttt{read})$, because the scalar transition map at time 2 is the same. At time 3 the current input token is the common symbol $\texttt{read}$, and the simulator's state is the same in both executions, so the produced output at time 3 must also be the same. This contradicts exact simulation, because $U_{w,p}$ outputs $T(x)$ in the first execution and $T'(x)$ in the second.

Hence the number of affine self-maps of $R_p^w$ cannot exceed the number of scalar affine self-maps of $R_{pw}$. But $|\text{Aff}(R_p^w)| = |R_p|^{w^2+w} = 2^{p(w^2+w)}$, whereas $|\text{Aff}(R_{pw})| = |R_{pw}|^2 = 2^{2pw}$. For every $w \geq 2$ we have $w^2 + w > 2w$, so $2^{p(w^2+w)} > 2^{2pw}$, a contradiction. Therefore no such simulator $V$ exists. $\qquad\square$

*Proof of Theorem 6.* Consider the following width-1, precision-3 machine $C$ over the input alphabet $X := \mathbb{Z}/8\mathbb{Z} \sqcup \{\texttt{inc}\}$. On the first input token $s \in \mathbb{Z}/8\mathbb{Z}$, the machine loads the hidden state with $s$. On every subsequent token $\texttt{inc}$, it updates

$$h \longmapsto h + 1 \pmod 8$$

and outputs the new state.

Assume, for contradiction, that there exists a width-3, precision-1 one-layer affine-state simulator $D$ over the state space $V := \mathbb{F}_2^3$. Let $e(s) \in V$ be the simulator state after reading the first token $s$. Since exact simulation must hold for all future suffixes of $\texttt{inc}$ tokens, the map $s \mapsto e(s)$ is injective, because both sets have size 8, it is bijective.

Let

$$f(x) = Ax + b \qquad (A \in \text{GL}(3, 2), \ b \in \mathbb{F}_2^3)$$

be the affine self-map of $V$ used by the simulator on the token $\texttt{inc}$. Fix any $s \in \mathbb{Z}/8\mathbb{Z}$. Exact simulation implies that after $k$ successive $\texttt{inc}$ tokens, the output must be $s + k \pmod 8$. In particular, for $k = 1, 2, \ldots, 8$ these outputs are pairwise distinct. Since the current input token is the same in each of these steps, equal simulator states would force equal outputs. Therefore the states $f(e(s))$, $f^2(e(s))$, $\ldots$, $f^8(e(s))$ are pairwise distinct. As $V$ has exactly 8 elements, $f$ must be a permutation of $V$ with a single orbit of length 8, and equivalently, $f$ must have order 8.

We now show that no affine permutation of $\mathbb{F}_2^3$ has order 8, which yields the contradiction.

**Lemma 3.** *If $f(x) = Ax + b$ is an affine permutation of $\mathbb{F}_2^3$, then the order of $f$ is not equal to 8.*

*Proof.* Write $r := \text{ord}(A)$. Since

$$f^m(x) = A^m x + \sum_{i=0}^{m-1} A^i b,$$

we obtain

$$f^r(x) = x + c, \qquad c := \sum_{i=0}^{r-1} A^i b.$$

Because $V = \mathbb{F}_2^3$ has characteristic 2, every translation $x \mapsto x + c$ has order 1 or 2. Hence the order of $f$ divides $2r$.

It remains to understand the possible orders of $A \in \text{GL}(3, 2)$. The characteristic polynomial of such an $A$ has degree 3 and nonzero constant term, so its irreducible factors over $\mathbb{F}_2$ are among

$$x + 1, \qquad x^2 + x + 1, \qquad x^3 + x + 1, \qquad x^3 + x^2 + 1.$$

The latter two cubic polynomials are primitive, hence contribute order 7, the quadratic polynomial contributes order 3, and the unipotent factor $(x+1)^m$ contributes 2-power order at most 4 in dimension 3. Therefore, $r \in \{1, 2, 3, 4, 7\}$. If $r \in \{1, 2, 3, 7\}$, then every divisor of $2r$ belongs to $\{1, 2, 3, 4, 6, 7, 14\}$, so in particular is not 8.

The only remaining possibility is $r = 4$. In that case $A$ is unipotent of index 3, so we may write $A = I + N$ with $N^3 = 0$. Then in characteristic 2,

$$A^2 = (I + N)^2 = I + N^2, \qquad A^3 = (I + N)^3 = I + N + N^2,$$

and therefore

$$I + A + A^2 + A^3 = I + (I + N) + (I + N^2) + (I + N + N^2) = 0.$$

Consequently,

$$c = (I + A + A^2 + A^3)b = 0,$$

so $f^4 = \mathrm{id}$. Thus in the case $r = 4$ the order of $f$ divides 4, again not 8.

Therefore no affine permutation of $\mathbb{F}_2^3$ has order 8. $\qquad\square$

By Lemma 3, the simulator transition on `inc` cannot have order 8, contradicting the previous paragraph. Hence no such simulator exists. $\qquad\square$

*Proof of Proposition 2.* For part (a), apply Theorem 4(A) with state dimension $d = w$. The given $L$-layer online-CoT SSM is simulated by a deterministic one-pass streaming algorithm using $O(Lwp + \log T_n)$ bits of persistent memory. Now apply Theorem 4(B) to this streaming algorithm with target state dimension $d = 1$ and precision $p' = O(Lwp + \log T_n)$. This yields a deterministic single-layer generalized SSM with online CoT, width 1, and precision $p'$ that simulates the original machine. By the construction in Theorem 4(B), the simulation uses one additional internal step per exogenous token.

For part (b), apply Theorem 4(A) to the given single-layer online-CoT SSM of width 1 and precision $q$. This produces a deterministic one-pass streaming algorithm using

$$O(q + \log T_n)$$

bits of persistent memory. Choose $C > 0$ sufficiently large to dominate the hidden constant in this bound. If $wp \geq C(q + \log T_n)$, then Theorem 4(B), applied with target state dimension $d = w$ and precision $p$, yields a deterministic single-layer generalized SSM with online CoT, width $w$, and precision $p$ that simulates the original machine. The final sentence follows by combining parts (a) and (b). $\qquad\square$

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
