# OpenReview forum: "On the Expressive Power and Limitations of Multi-Layer SSMs"
_TMLR — Under review for TMLR_

### Review · Reviewer_tBMY · 2026-05-03

**Summary Of Contributions:**

**Summary**.
This paper analyzes the expressive power of multi-layer state-space models (SSMs), focusing on their ability to solve compositional tasks and how this depends on depth, memory, and precision. The authors establish theoretical limits showing that shallow SSMs struggle with function composition, while increased depth enables strictly greater computational power, yielding a depth hierarchy. A key contribution is the distinction between offline and online chain-of-thought (CoT): offline CoT does not improve expressiveness, whereas online CoT significantly enhances capability, making SSMs comparable to streaming algorithms. The paper further examines the tradeoff between width and precision, showing they are not interchangeable in standard SSMs but become equivalent under online CoT. Overall, the work provides a unified theoretical perspective on how architectural and computational factors govern the capabilities of SSMs.

**Strengths**
* The manuscript is written very clearly with respect to its scope and structure. In particular, I appreciated the overview on page two summarizing the main conclusions and providing signposts to the relevant sections.
* The discussion of related work is solid. I appreciate the clarity in how the paper is positioned within prior work and what the precise research contribution is.
* As noted in the section below, the work is both interesting and relevant. SSMs are used across a wide range of tasks, and CoT has emerged as an important principle for reasoning models. An analysis of the capabilities of SSMs (in conjunction with CoT) therefore constitutes a meaningful research contribution.
* The mathematical notation is clean, and the claims (primarily stated as theorems and lemmas) appear to be supported by rigorous formal proofs.

**Weaknesses**
* This is not a major weakness, but to provide a complete picture: while I highlighted clarity and structure as strengths, I still found parts of the proofs difficult to follow. I attribute this more to my own limited background in this area than to issues in presentation. Nevertheless, the highly theoretical nature of the work may limit its accessibility to a broader audience.
* Minor: While not strictly necessary for this type of work, the practical implications could be stated more explicitly to make the findings accessible to a broader audience. (Q1): Could the authors formulate clearer takeaways or “call-to-actions” for practitioners? In this context, the use of the terms “offline” and “online” CoT might be reconsidered. While they are well introduced and explained in the manuscript, they do not appear to be standard terminology. If a practitioner-oriented summary is included, it would be beneficial to either avoid these terms to ensure that this section is fully self-explanatory without requiring detailed context from the paper.

**Audience:**

Yes

**Audience Explanation:**

SSMs are widely use for a variety of different tasks. Thus, theoretical foundation work wrt their expressiveness is interesting and relevant. The concept of CoTs has been proven successful and relevant for transformer-based reasoning models. An analysis about its effectiveness for SSMs is relevant and desireable.

**Broader Impact Concerns:**

No concerns.

**Claims And Evidence:**

Yes

**Claims Explanation:**

While I am not an expert in this area, the claims (primarily stated as theorems and lemmas) appear to be well supported by formal proofs.

**Requested Changes:**

Please see **Q1**. I do not consider this point critical for acceptance.

---

> ### Author Response · Authors · 2026-07-16
> **Response to Reviewer tBMY**
>
> We thank the reviewer for the careful and encouraging assessment. We agree that the practical interpretation and proof-level accessibility can be improved.
>
> **Terminology**
>
> We agree that “offline/online CoT” is not standard terminology. Our distinction is operational: *post-input reasoning* generates all self-generated tokens after the final input token, whereas *input-interleaved reasoning* may insert such tokens between input tokens. We will use these descriptive phrases in the abstract, introduction, and practitioner summary, retaining “offline/online CoT” only as short aliases after the formal definitions in Section 6. We will also clarify that “online” refers to token timing, not online learning.
>
> **Practical takeaways and scope**
>
> We will consolidate the discussion in Section 8 into a concise practitioner-facing summary:
>
> 1. The depth lower bound concerns a controlled sequential-composition benchmark. It does not imply that every practical task reduces to function composition. A direct empirical test is to vary the composition length relative to the SSM depth.
>
> 2. In our generalized model, post-input computation cannot circumvent the information bottleneck used by our communication reduction, whereas input-interleaved computation gives the model the power of a general one-pass streaming algorithm within the deterministic setting analyzed by Theorem 4.
>
> 3. This additional expressiveness is not free. The construction uses one internal token per input token, equivalently, two generalized SSM steps per input. Experiments should therefore report internal token, latency, and throughput overhead.
>
> 4. Equal total storage does not make state dimension and numerical precision universally interchangeable, so they should be treated as separate ablation axes.
>
> We will emphasize that these are results on expressiveness and space, not guarantees of learnability, accuracy, or wall-clock efficiency. Evaluating pause/thinking-token mechanisms and their accuracy-cost tradeoff remains an empirical direction, and we hope that, on the basis of this theoretical work, such experimental studies will be conducted.
>
> **Proof accessibility**
>
> We will add local proof roadmaps: Section 4 will summarize the affine summary, forward protocol, serialization, and pointer-chasing steps. Before Theorem 4, we will explain the two simulations operationally. The main text will retain theorem statements and proof ideas, while detailed serialization and algebraic casework will move to appendices.
>
> We believe these changes will make the scope and practical implications substantially easier to understand.

---

### Review · Reviewer_qggd · 2026-05-23

**Summary Of Contributions:**

### Contributions

This work provides an extensive theoretical analysis on the expressive power of multi-layer SSMs. In particular, they show that (a) any $L$-layer SSM solving the $(L+3)$-function composition problem must satisfy $d^2p=\Omega(N/L^3)$, where $d$, $p$, and $N$ are state dimension, per-scalar precision, and problem size, respectively; (b) online CoT, unlike offline CoT, can circumvent the previously obtained lower bound; and (c) width and precision are not interchangeable in a non-CoT SSM, whereas they are interchangeable in the online CoT SSM.

### Strengths

* The paper introduces a novel technique of connecting multi-layer SSMs to multi-party communication via a forward communication model.
* This is the first work that provides a formal analysis of the CoT for SSMs.

### Weaknesses

The connection between the theoretical analysis presented in this paper and the practical usage of SSMs is unclear. Particularly:
* $K$-function composition problem is nowhere near what SSMs are usually trained for in practice. Can an arbitrary problem be reformulated into a function composition problem? How general is the equivalence class?
* It is questionable whether online CoT could be implemented in practice without hurting the computational efficiency of the base model by a huge extent.

**Audience:**

Yes

**Audience Explanation:**

SSM is an active and competitive research area, so there will definitely be individuals in TMLR's audience that would be interested in the theoretical work on SSM's fundamental characteristics.

**Broader Impact Concerns:**

I do not have any ethical concerns specific to this work, and I do not believe a Broader Impact Statement is strictly necessary, though I note that the paper does not include one.

**Claims And Evidence:**

Yes

**Claims Explanation:**

This is a theory paper and the claims are supported by the proofs.

**Requested Changes:**

All the requested changes are non-critical.

1. The theory presented in the paper mainly focuses on $K$-function composition problem. However, such problem class is drastically different from real-world SSM use cases. The authors should explain why $K$-function composition problem can be regarded as a representative problem class.

2. The authors should discuss the feasibility of online CoT. It is questionable whether they could be implemented (at least approximately) without degrading the computational complexity of the base model by a great extent.

3. Please consider moving the proofs to the appendix and use the freed-up space to discuss the meanings and consequences of each theoretical findings more thoroughly.

### Miscellaneous comments

1. The first sentence of the second paragraph of the "Techniques" section on page 2 is difficult to understand: For the CoT results, ... stream is decisive.

2. The notation $PC(N, K)$ is used without definition.

---

> ### Author Response · Authors · 2026-07-16
> **Response to Reviewer qggd**
>
> We thank the reviewer for the positive assessment and agree that the theory's practical scope should be clearer.
>
> **1. Why is function composition representative?**
>
> Function composition captures the basic operation of carrying the output of one computation into the next. We do not claim that every compositional task inherits the same lower bound. Its significance lies in being a canonical complete-type task for sequential information propagation: related structures arise in iterated state transitions, pointer chasing, repeated transformations, and multi-step algorithmic reasoning. Thus, the result identifies a broader architectural obstruction, information cannot always be propagated through an arbitrarily long adaptive chain with a fixed layer budget, while the precise quantitative statement remains limited to our formal model and problem family.
>
> An arbitrary staged computation can be represented abstractly as a composition of transition maps, but converting it to our explicit-table benchmark may require exponentially large tables or fail to preserve the required stream order. Therefore, we make no universal reduction claim. We will add this discussion after Definition 2, describe function composition as a canonical stress test rather than a model of every practical task, and explicitly state the blockwise encoding required by the theorem. We will also clarify that our lower bounds apply to subclasses covered by Definition 1, whereas the positive constructions for the generalized model do not automatically establish realizability or learnability in a specific S4/Mamba architecture.
>
> **2. Feasibility and cost of input-interleaved CoT**
>
> We agree that Theorem 4 is an expressiveness and memory result, not a claim that interleaved CoT is computationally free. If $k_i$ thought tokens are generated after $x_i$, the processed length is
> $$
> T=n+\sum_{i=1}^{n}k_i,
> $$
> so the model performs $T$ recurrent updates. The unrestricted definition, therefore, provides no general runtime bound.
>
> Architecturally, within our model, interleaved CoT can be implemented by pausing the external stream after processing $x_i$, applying the ordinary SSM update to one or more self-generated thought tokens, and then resuming with $x_{i+1}$. A generated token is fed back through the usual autoregressive interface. In Corollary 2, exactly one thought token is inserted after every exogenous token, so no adaptive stopping mechanism is required and the augmented stream has length $2n$.
>
> Thus, the construction has a factor-two update-count overhead and remains linear under generalized-step accounting. This is not necessarily a factor-two wall-clock overhead: adaptive feedback can sacrifice associative-scan parallelism, and the theorem does not bound the implementation cost of the transition, readout, or token-generation maps. We will state this work-memory-latency distinction explicitly and present practical pause/thinking-token implementations as an empirical direction rather than an efficiency guarantee.
>
> **3. Proof organization**
>
> We agree. The main text will retain theorem statements, central invariants, and short proof roadmaps. Detailed protocol serialization, parameter-level constructions, and algebraic casework from Sections 4-7 will move to appendices. We will use the released space for benchmark scope, CoT cost accounting, and practitioner-facing interpretations.
>
> **4. Minor comments**
>
> We will replace the difficult sentence on page 2 with:
>
> “For the CoT results, the decisive issue is when thought tokens are generated relative to the exogenous input stream.”
>
> Definition 4 defines $\mathrm{PC}_k$, so we will also replace the undefined $\mathrm{PC}(N,K)$ by $\mathrm{PC}_K$ throughout.